# RefLoRA: Refactored Low-Rank Adaptation for Efficient Fine-Tuning of Large Models

**Yilang Zhang**
Department of ECE
University of Minnesota
Minneapolis, MN 55414
zhan7453@umn.edu

**Bingcong Li**
Department of CS
ETH Zürich
8092 Zürich, Switzerland
bingcong.li@inf.ethz.ch

**Georgios B. Giannakis**
Department of ECE
University of Minnesota
Minneapolis, MN 55414
georgios@umn.edu

## Abstract

Low-Rank Adaptation (LoRA) lowers the computational and memory overhead of fine-tuning large models by updating a low-dimensional subspace of the pre-trained weight matrix. Albeit efficient, LoRA exhibits suboptimal convergence and noticeable performance degradation, due to inconsistent and imbalanced weight updates induced by its nonunique low-rank factorizations. To overcome these limitations, this article identifies the optimal low-rank factorization per step that minimizes an upper bound on the loss. The resultant refactored low-rank adaptation (RefLoRA) method promotes a flatter loss landscape, along with consistent and balanced weight updates, thus speeding up stable convergence. Extensive experiments evaluate RefLoRA on natural language understanding, and common-sense reasoning tasks with popular large language models including DeBERTaV3, LLaMA-7B, LLaMA2-7B and LLaMA3-8B. The numerical tests corroborate that RefLoRA converges faster, outperforms various benchmarks, and enjoys negligible computational overhead compared to state-of-the-art LoRA variants.

## 1 Introduction

Large language models (LLMs) have revolutionized a wide spectrum of applications including chatbots [1], code generation [8], and scientific discovery [55]. Despite their success, adapting LLMs to specific tasks remains computationally demanding. LLMs are built on a two-stage learning process, namely *pre-training* and *fine-tuning*. Pre-training is performed on massive, Internet-scale corpora. This endows LLMs with in-text comprehension and generation abilities, but results in models with billions to trillions parameters [1, 17]. Though broad language capabilities are granted, pre-trained LLMs are yet not tailored for specialized applications. To acquire domain-specific expertise, LLMs must be further trained on downstream tasks, what is known as fine-tuning. With the continually growing model size however, conventional full fine-tuning approaches (optimizing all model parameters) can be increasingly prohibitive due to the immense GPU memory and substantial computational capacity demands, rendering them impossible for individual users and organizations.

To tackle the computational bottleneck, parameter-efficient fine-tuning (PEFT) [22] has been investigated to enhance fine-tuning efficiency. As opposed to fully fine-tuning all parameters, PEFT methods either optimize a sparse subset [18, 54], or introduce additional lightweight trainable parameters while keeping the pre-trained ones frozen [22, 46, 36, 29, 33]. Among these approaches, low-rank adaptation (LoRA) [23] has gained popularity due to its low additional cost during inference. LoRA presumes that the parameter updates during fine-tuning lie on a low-dimensional manifold, and thus can be captured by a low-rank matrix. Though effective, LoRA is observed to suffer from challenges such as slow and unstable convergence [41], inconsistent and unbalanced weight updates [67, 20], and notable performance gaps relative to full fine-tuning [23]. One key reason behind these challenges is the nonuniqueness of LoRA's low-rank factorization [67].

39th Conference on Neural Information Processing Systems (NeurIPS 2025).

To cope with these challenges, this work advocates "refactoring" low-rank adaption (RefLoRA), a novel approach that commits to the optimal factorization per step. Our contribution is threefold:

- We show that LoRA's inconsistent weight updates can be characterized by a symmetric positive definite matrix. RefLoRA dynamically selects the optimal one by minimizing an upper bound on the loss, resulting in flatter landscape of the loss that facilitates stable and efficient optimization.
- The optimal factorization is proven to have a closed-form global solution; to yield consistent and balanced weight updates; and to bring about a lower overhead compared to SOTA approaches. Moreover, a simplified variant termed RefLoRA-S is developed to further reduce complexity.
- Extensive numerical tests are conducted on matrix factorization, natural language understanding, and commonsense reasoning benchmarks with popular LLMs scaled up to 8B parameters, demonstrating RefLoRA's faster convergence and consistent performance gain.

**Related work.** Building upon LoRA, several have been developed to ameliorate its performance. Some revise vanilla LoRA's architecture or dynamically adjust its hyperparameters. For instance, DoRA [65] decomposes LoRA weights into magnitude and direction components to improve learning capacity and training stability. AdaLoRA [70] and GeoLoRA [51] allocates per-layer rank on-the-fly to prune less salient updates. FedPara [25] (a.k.a. LoHa) and LoKr [66] respectively integrate the low-rank structure with Hadamard and Kronecker products for reduced communication cost and improved model expressiveness. Another line of work boosts LoRA's behavior in early fine-tuning epochs through well-designed initialization. PiSSA [41] leverages truncated singular value decomposition (SVD) to keep the low-rank initialization close to the pre-trained weights, whereas LoRA-GA [60] aligns the first update direction with full fine-tuning. Other variants refine LoRA's per-step optimization to promote empirical convergence. LoRA-Pro [61] redirects LoRA's gradient to match the weight update of full fine-tuning. LoRA-RITE [67] substitutes the default Adam optimizer [26] with customized gradient calculation and moment estimation scheme. However, these gradient-altering strategies necessitate meticulous crafting to ensure stability and convergence. Our approach falls within the same family, but adheres strictly to standard backpropagation rule, avoiding direct gradient manipulation and requiring less computational overhead. To meet constraints of space limits, additional LoRA variants are discussed in Appendix A.

## 2 Preliminaries and nonunique low-rank factorizations

Consider a general weight matrix $\mathbf{W} \in \mathbb{R}^{m \times n}$ parameterizing a large model. With initialization being the pre-trained matrix $\mathbf{W}_0 := \mathbf{W}^{\mathrm{pt}}$ and $t$ indexing iteration, conventional full fine-tuning updates the sizable matrix $\mathbf{W}_t$ by backpropagating the loss function $\ell(\mathbf{W}_t)$. Though straightforward, this approach requires an excessive memory footprint and major computational cost.

Aiming at efficient fine-tuning, LoRA [23] freezes $\mathbf{W}^{\mathrm{pt}}$ and presumes that the weight increment exhibits a low-rank structure $\mathbf{W}_t = \mathbf{W}^{\mathrm{pt}} + \mathbf{A}_t \mathbf{B}_t^\top$, where $\mathbf{A}_t \in \mathbb{R}^{m \times r}$, $\mathbf{B}_t \in \mathbb{R}^{n \times r}$, and $r \ll \min\{m, n\}$ is a preselected constant. Consequently, LoRA's objective function boils down to

$$\min_{\mathbf{A}, \mathbf{B}} \mathcal{L}(\mathbf{A}, \mathbf{B}) := \ell(\mathbf{W}^{\mathrm{pt}} + \mathbf{A}\mathbf{B}^\top).$$

This reformulation reduces the number of trainable parameters to $(m + n)r \ll mn$, thus effectively minimizing the associated memory and computation costs. As for initialization, LoRA draws entries of $\mathbf{A}_0$ from a zero-mean normal distribution with small variance $\sigma^2$ for numerical stability; and $\mathbf{B}_0 = \mathbf{0}$ to ensure $\mathbf{W}_0 = \mathbf{W}^{\mathrm{pt}}$. This choice gives rise to imbalanced updates of $\mathbf{A}_t$ and $\mathbf{B}_t$, and decelerates empirical convergence especially in early epochs. At the first iteration for example, the chain rule implies that $\nabla_{\mathbf{A}_0} \mathcal{L}(\mathbf{A}_0, \mathbf{B}_0) = \nabla\ell(\mathbf{W}^{\mathrm{pt}} + \mathbf{A}_0 \mathbf{B}_0^\top)\mathbf{B}_0 = \mathbf{0}$, meaning no update to $\mathbf{A}_0$. In comparison, $\mathbf{B}_0$'s gradient $\nabla_{\mathbf{B}_0} \mathcal{L}(\mathbf{A}_0, \mathbf{B}_0) = \nabla\ell(\mathbf{W}^{\mathrm{pt}})^\top \mathbf{A}_0$ is generally non-zero. In fact, it turns out that $\|\mathbf{B}_t\|_{\mathrm{F}} \ll \|\mathbf{A}_t\|_{\mathrm{F}}$ and $\|\nabla_{\mathbf{A}_t} \mathcal{L}(\mathbf{A}_t, \mathbf{B}_t)\|_{\mathrm{F}} \ll \|\nabla_{\mathbf{B}_t} \mathcal{L}(\mathbf{A}_t, \mathbf{B}_t)\|_{\mathrm{F}}$ when $t$ is small [41], and these unbalanced low-rank factors and updates can markedly impede the convergence of LoRA.

Additionally, the *nonuniqueness* of LoRA's low-rank factors leads to inconsistent parameter updates. While this issue has been previously recognized [43, 67], its implications for optimization remain underexplored. Specifically, for any alternative decomposition $(\tilde{\mathbf{A}}_t, \tilde{\mathbf{B}}_t)$ satisfying $\tilde{\mathbf{A}}_t \tilde{\mathbf{B}}_t^\top = \mathbf{A}_t \mathbf{B}_t^\top$, the forward pass and loss remain intact, whereas the update of $\mathbf{W}_t$ can differ significantly. For illustration, consider standard gradient descent (GD) iterations

$$\mathbf{A}_{t+1} = \mathbf{A}_t + \Delta\mathbf{A}_t := \mathbf{A}_t - \eta\nabla\ell(\mathbf{W}_t)\mathbf{B}_t, \quad \mathbf{B}_{t+1} = \mathbf{B}_t + \Delta\mathbf{B}_t := \mathbf{B}_t - \eta\nabla\ell(\mathbf{W}_t)^\top \mathbf{A}_t \quad (1)$$

where $\eta > 0$ denotes the learning rate. The corresponding update to the weight matrix is

$$\Delta \mathbf{W}_t := \mathbf{W}_{t+1} - \mathbf{W}_t = \mathbf{A}_{t+1}\mathbf{B}_{t+1}^\top - \mathbf{A}_t\mathbf{B}_t^\top = \mathbf{A}_t\Delta\mathbf{B}_t^\top + \Delta\mathbf{B}_t\mathbf{A}_t^\top + \Delta\mathbf{A}_t\Delta\mathbf{B}_t^\top. \quad (2)$$

Alternatively, GD can be also performed with the equivalent pair $(\tilde{\mathbf{A}}_t, \tilde{\mathbf{B}}_t)$, yielding

$$\mathbf{A}_{t+1} = \tilde{\mathbf{A}}_t + \Delta\tilde{\mathbf{A}}_t := \tilde{\mathbf{A}}_t - \eta\nabla\ell(\mathbf{W}_t)\tilde{\mathbf{B}}_t, \quad \mathbf{B}_{t+1} = \tilde{\mathbf{B}}_t + \Delta\tilde{\mathbf{B}}_t := \tilde{\mathbf{B}}_t - \eta\nabla\ell(\mathbf{W}_t)^\top\tilde{\mathbf{A}}_t. \quad (3)$$

The resultant parameter update is thereby $\Delta\tilde{\mathbf{W}}_t := \tilde{\mathbf{A}}_t\Delta\tilde{\mathbf{B}}_t^\top + \Delta\tilde{\mathbf{B}}_t\tilde{\mathbf{A}}_t^\top + \Delta\tilde{\mathbf{A}}_t\Delta\tilde{\mathbf{B}}_t^\top$, which can remarkably deviate from $\Delta\mathbf{W}_t$ in (2), despite both factorizations representing the same $\mathbf{W}_t$. Further elaboration on this issue will be provided in the ensuing section.

The following notational conventions are adopted throughout the paper.

**Notation.** Bold lowercase (capital) letters denote vectors (matrices); $\|\cdot\|_2$, $\|\cdot\|_*$, $\|\cdot\|_F$ and $\langle\cdot,\cdot\rangle_F$ stand for $\ell_2$-, nuclear-, Frobenius-norm and Frobenius inner product; $\mathrm{Col}(\cdot)$, $\mathrm{Null}(\cdot)$ and $[\cdot]_i$ represent column space, null space and the $i$-th entry/column of a vector/matrix; $\cdot^\dagger$, $\mathrm{tr}(\cdot)$, and $\mathrm{rank}(\cdot)$ refer to the Moore-Penrose pseudoinverse, trace, and rank; $\lambda_i(\cdot)$ and $\sigma_i(\cdot)$ are the $i$-th largest eigenvalue and singular value; $\mathbb{S}_{++}^r$ indicates the set of $r \times r$ symmetric positive definite (SPD) matrices; $\mathrm{GL}(r)$ denotes the general linear group of degree $r$ (i.e., the set of $r \times r$ invertible matrices); and $\mathrm{O}(r)$ stands for the orthogonal group of degree $r$ (i.e., the set of $r \times r$ orthogonal matrices).

## 3 Low-rank adaptation with optimal refactoring

This section delves into the nonuniqueness of LoRA's factorization, and identifies the optimal one that minimizes the loss $\ell(\mathbf{W}_t + \Delta\tilde{\mathbf{W}}_t)$. It is first demonstrated that *all* possible factors $(\tilde{\mathbf{A}}_t, \tilde{\mathbf{B}}_t)$ can be characterized by an invertible matrix $\mathbf{P}_t$, and the weight update $\Delta\tilde{\mathbf{W}}_t$ is fundamentally governed by an SPD matrix $\mathbf{S}_t := \mathbf{P}_t\mathbf{P}_t^\top$. Then, we will derive an upper bound of $\ell(\mathbf{W}_t + \Delta\tilde{\mathbf{W}}_t)$ as a function of $\mathbf{S}_t$, whose global minimum will be obtained in closed form. Building on these theoretical insights, we will optimally refactor LoRA's low-rank matrices per iteration to obtain more effective updates, that are at the center of our "refactored" low-rank adaptation (RefLoRA) approach. All proofs in this section are deferred to Appendix B.

### 3.1 Characterizing LoRA's factorization and weight update

Our analysis begins with the following mild assumption, which has been utilized and validated on various realistic datasets [61].

**Assumption 1.** $\mathrm{rank}(\mathbf{A}_t) = \mathrm{rank}(\mathbf{B}_t) = r, \forall t > 0$.

Assumption 1 asserts that the tall matrices $\mathbf{A}_t$ and $\mathbf{B}_t$ maintain full column rank after the first iteration. Since LoRA seeks to approximate the full update of $\mathbf{W}_t$ using a low-rank matrix, this assumption essentially reflects the effectiveness of LoRA's parameterization. Under Assumption 1, the next lemma reveals that *all* equivalent factorizations can be captured by an $r \times r$ invertible matrix $\mathbf{P}_t$.

**Lemma 1.** *With Assumption 1 in effect, it holds that*

$$\{(\tilde{\mathbf{A}}_t, \tilde{\mathbf{B}}_t) \mid \tilde{\mathbf{A}}_t\tilde{\mathbf{B}}_t^\top = \mathbf{A}_t\mathbf{B}_t^\top\} = \{(\mathbf{A}_t\mathbf{P}_t, \mathbf{B}_t\mathbf{P}_t^{-\top}) \mid \mathbf{P}_t \in \mathrm{GL}(r)\}. \quad (4)$$

*Moreover, if $\mathbf{P}_t \in \mathrm{O}(r)$, then $\Delta\tilde{\mathbf{W}}_t = \Delta\mathbf{W}_t$.*

Beyond characterizing the structure of equivalent factorizations, Lemma 1 implies that consistent weight updates are preserved when $(\tilde{\mathbf{A}}_t, \tilde{\mathbf{B}}_t)$ differs from $(\mathbf{A}_t, \mathbf{B}_t)$ up to a rotation and reflection. This naturally raises the question: *which factorization, or $\mathbf{P}_t$, is preferable for effective optimization?*

To answer the last question, we first present an important observation about $\Delta\tilde{\mathbf{W}}_t$. When setting $\tilde{\mathbf{A}}_t = \mathbf{A}_t\mathbf{P}_t$ and $\tilde{\mathbf{B}}_t = \mathbf{B}_t\mathbf{P}_t^{-\top}$, it can be readily verified (see (14) of Appendix B.1) that

$$\Delta\tilde{\mathbf{W}}_t = \mathbf{A}_t(\mathbf{P}_t\mathbf{P}_t^\top)\Delta\mathbf{B}_t^\top + \Delta\mathbf{A}_t(\mathbf{P}_t\mathbf{P}_t^\top)^{-1}\mathbf{B}_t^\top + \Delta\mathbf{A}_t\Delta\mathbf{B}_t^\top. \quad (5)$$

Letting $\mathbf{P}_t = \mathbf{U}_t^P\mathbf{\Sigma}_t^P\mathbf{V}_t^{P\top}$ denote the SVD of $\mathbf{P}_t$, it is clear that $\Delta\tilde{\mathbf{W}}_t$ is fully determined by the $r \times r$ SPD matrix $\mathbf{S}_t := \mathbf{P}_t\mathbf{P}_t^\top = \mathbf{U}_t^P(\mathbf{\Sigma}_t^P)^2\mathbf{U}_t^{P\top} \in \mathbb{S}_{++}^r$. This implies that $\mathbf{V}_t^P$ can be chosen arbitrarily from $\mathrm{O}(r)$. In other words, $\mathbf{P}_t$ can be right-multiplied by any orthogonal matrix, without affecting $\Delta\tilde{\mathbf{W}}_t$. This observation agrees with the last statement of Lemma 1 by replacing $(\mathbf{A}_t, \mathbf{B}_t)$ and $\mathbf{P}_t$ with $(\mathbf{A}_t\mathbf{U}_t^P\mathbf{\Sigma}_t^P, \mathbf{B}_t\mathbf{U}_t^P(\mathbf{\Sigma}_t^P)^{-1})$ and $\mathbf{V}_t^P$. Consequently, identifying the optimal factorization boils down to selecting an ideal $\mathbf{S}_t \in \mathbb{S}_{++}^r$.

## 3.2 Optimizing $\mathbf{S}_t$ via loss upper bound minimization

Our key idea is to select the $\mathbf{S}_t$ that minimizes the loss $\ell(\mathbf{W}_{t+1}) = \ell(\mathbf{W}_t + \Delta\tilde{\mathbf{W}}_t(\mathbf{S}_t))$. Unfortunately, directly minimizing this objective over $\mathbf{S}_t$ requires exhaustive search due to the model nonlinearity, which is infeasible in practice. As an alternative, we derive a tractable upper bound on $\ell(\mathbf{W}_t + \Delta\tilde{\mathbf{W}}_t(\mathbf{S}_t))$, and minimize the bound to obtain an optimal $\mathbf{S}_t$. Our motivation stems from GD, which relies on the following Lipschitz smoothness assumption.

**Assumption 2.** *The loss function $\ell$ has $L$-Lipschitz gradient; i.e., $\|\nabla\ell(\mathbf{W}) - \nabla\ell(\mathbf{W}')\|_{\mathrm{F}} \leq L\|\mathbf{W} - \mathbf{W}'\|_{\mathrm{F}}, \ \forall \mathbf{W}, \mathbf{W}' \in \mathbb{R}^{m\times n}$.*

Assumption 2 is equivalent to requiring Lipschitz smoothness of $\ell$ w.r.t. the vectorized weight matrix $\mathrm{vec}(\mathbf{W})$, which is fairly mild and common in machine learning [52, 16] and optimization [3, 6]. Under this assumption, the loss admits the following quadratic upper bound

$$\ell(\mathbf{W}_t + \Delta\mathbf{W}_t) \leq \ell(\mathbf{W}_t) + \langle\nabla\ell(\mathbf{W}_t), \Delta\mathbf{W}_t\rangle_{\mathrm{F}} + \frac{L}{2}\|\Delta\mathbf{W}_t\|_{\mathrm{F}}^2. \tag{6}$$

Minimizing the bound yields the optimum $\Delta\mathbf{W}_t^* = -\frac{1}{L}\nabla\ell(\mathbf{W}_t)$, thus recovering the standard GD used in full fine-tuning. Given that $L$ is typically unknown in practice, the optimal learning rate $1/L$ is replaced by a hyperparameter, whose value can be tuned via grid search on a validation dataset.

Although $\ell$ is often Lipschitz smooth w.r.t. $\mathbf{W}_t$, its smoothness constants w.r.t. $\mathbf{A}_t$ and $\mathbf{B}_t$ can be unbounded due to the bilinear structure, unless one assumes boundedness or convergence of $\mathbf{A}_t$ and $\mathbf{B}_t$ [14]. Since these two assumptions are overly restrictive, they will be avoided in our analysis.

The nonlinear dependence of $\Delta\tilde{\mathbf{W}}_t$ on $\mathbf{S}_t$ (cf. (5)) prevents an analytical solution when directly optimizing $\mathbf{S}_t$ over the quadratic upper bound of $\ell(\mathbf{W}_t + \Delta\tilde{\mathbf{W}}_t(\mathbf{S}_t))$. This motivates iterative solvers involving matrix multiplication with the sizable matrix $\nabla\ell(\mathbf{W}_t) \in \mathbb{R}^{m\times n}$. To mitigate the overhead, the next proposition relaxes the quadratic upper bound to factor out $\|\nabla\ell(\mathbf{W}_t)\|_2^2$, thus decoupling $\mathbf{S}_t$'s optimization from $\nabla\ell(\mathbf{W}_t)$.

**Proposition 2.** *Consider GD update (3) with $\tilde{\mathbf{A}}_t = \mathbf{A}_t\mathbf{P}_t$, $\tilde{\mathbf{B}}_t = \mathbf{B}_t\mathbf{P}_t^{-\top}$, and $\mathbf{S}_t := \mathbf{P}_t\mathbf{P}_t^{\top}$. Under Assumptions 1 and 2, it follows that*

$$\ell(\mathbf{W}_t + \Delta\tilde{\mathbf{W}}_t(\mathbf{S}_t)) \leq \frac{L\eta^2}{2}\|\nabla\ell(\mathbf{W}_t)\|_2^2\left(\|\mathbf{A}_t\mathbf{S}_t^{\frac{1}{2}}\|_{\mathrm{F}}^2 + \|\mathbf{B}_t\mathbf{S}_t^{-\frac{1}{2}}\|_{\mathrm{F}}^2 - \frac{1}{L\eta}\right)^2 + \mathcal{O}(L\eta^3) + \mathrm{Const.} \tag{7}$$

*where* Const. *refers to constants that do not rely on $\mathbf{S}_t$.*

The high-order term $\mathcal{O}(L\eta^3)$ originates from $\Delta\mathbf{A}_t\Delta\mathbf{B}_t^{\top}$ in (5). As $\eta$ is typically small ($\sim \mathcal{O}(10^{-4})$), this term is negligible in practice [60, 67]. As a consequence, the upper bound in (7) is dominated by its first term. This leads to our RefLoRA objective

$$\min_{\mathbf{S}_t\in\mathbb{S}_{++}^r}\left(\|\mathbf{A}_t\mathbf{S}_t^{\frac{1}{2}}\|_{\mathrm{F}}^2 + \|\mathbf{B}_t\mathbf{S}_t^{-\frac{1}{2}}\|_{\mathrm{F}}^2 - \frac{1}{L\eta}\right)^2. \tag{8}$$

Consider for convenience the variables $\tilde{\mathbf{S}}_t$ and $\tilde{C}_t$ defined as

$$\tilde{\mathbf{S}}_t := (\mathbf{A}_t^{\top}\mathbf{A}_t)^{-\frac{1}{2}}\left[(\mathbf{A}_t^{\top}\mathbf{A}_t)^{\frac{1}{2}}\mathbf{B}_t^{\top}\mathbf{B}_t(\mathbf{A}_t^{\top}\mathbf{A}_t)^{\frac{1}{2}}\right]^{\frac{1}{2}}(\mathbf{A}_t^{\top}\mathbf{A}_t)^{-\frac{1}{2}}, \quad \tilde{C}_t := 2\|\mathbf{A}_t\mathbf{B}_t^{\top}\|_* \tag{9}$$

based on which (8) can be solved in closed form as established in the following theorem.

**Theorem 3.** *Under Assumptions 1 and 2, the global optimum of (8) satisfies*

$$\mathbf{S}_t^*\begin{cases}= \tilde{\mathbf{S}}_t, & \text{if } \eta \geq \frac{1}{\tilde{C}_tL} \text{ or } \eta < 0 \\ \ni \left[(\tilde{C}_tL\eta)^{-1} \pm \sqrt{(\tilde{C}_tL\eta)^{-2} - 1}\right]\tilde{\mathbf{S}}_t, & \text{if } 0 < \eta < \frac{1}{\tilde{C}_tL}\end{cases}. \tag{10}$$

Theorem 3 states that if $\eta > 0$ is not too small, then (8) admits a unique global optimum $\tilde{\mathbf{S}}_t$. Otherwise, there can be multiple optima, while two can always be constructed by appropriately scaling $\tilde{\mathbf{S}}_t$. In addition, $\eta < 0$ is included in (10) for visualization purposes. We remark that $\tilde{\mathbf{S}}_t$ in (9) is known as the matrix geometric mean $(\mathbf{A}_t^{\top}\mathbf{A}_t)^{-1}\#(\mathbf{B}_t^{\top}\mathbf{B}_t)$ of $(\mathbf{A}_t^{\top}\mathbf{A}_t)^{-1}$ and $(\mathbf{B}_t^{\top}\mathbf{B}_t)$ [32], which can be also written as $\tilde{\mathbf{S}}_t = (\mathbf{A}_t^{\top}\mathbf{A}_t)^{-1}[\mathbf{A}_t^{\top}\mathbf{A}_t\mathbf{B}_t^{\top}\mathbf{B}_t]^{\frac{1}{2}}$; cf. Lemma 7 in the Appendix.

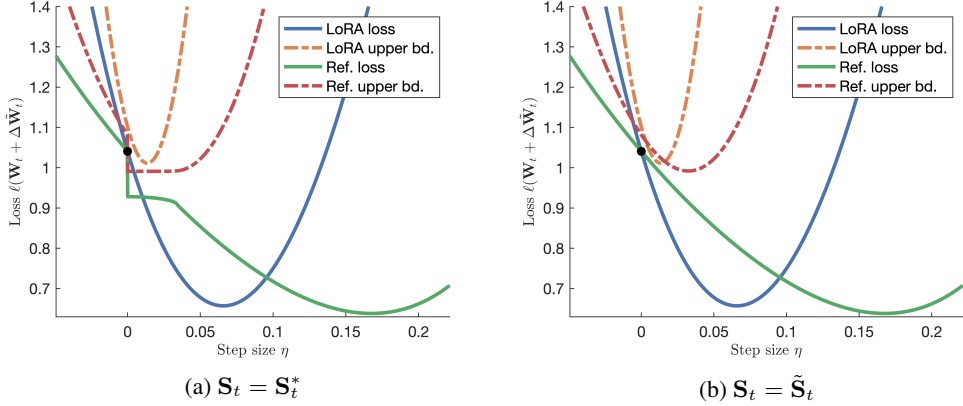

Figure 1: Visualization of loss $\ell(\mathbf{W}_t + \Delta\tilde{\mathbf{W}}_t)$ and upper bound (7). LoRA corresponds to $\mathbf{S}_t = \mathbf{I}_r$, while our refactoring (ref.) optimizes $\mathbf{S}_t$.

Figure 1 plots the loss $\ell(\mathbf{W}_t + \Delta\tilde{\mathbf{W}}_t)$, and the upper bound (7) of a numerical example as a function of $\eta$; see also Appendix D.2 for details. LoRA corresponds to the non-optimized $\mathbf{S}_t = \mathbf{I}_r$, whereas our refactoring selects $\mathbf{S}_t = \mathbf{S}_t^*$ based on Theorem 3. Notably, $\eta = 0$ leads to a jump discontinuity for our refactoring. Figure 1a shows that by optimizing the upper bound (7), the associated loss becomes lower and flatter. This enables a larger step size $\eta$ to achieve a lower loss, thereby accelerating the empirical convergence of LoRA; see Section 4 for experiments corroborating this claim.

The flatter loss landscape arises thanks to balancing $\tilde{\mathbf{A}}_t$ and $\tilde{\mathbf{B}}_t$. Indeed, the imbalance of $\mathbf{A}_t$ and $\mathbf{B}_t$ in vanilla LoRA necessitates different step sizes [20]. Appendix B.3 proves that the balance $\tilde{\mathbf{A}}_t^\top \tilde{\mathbf{A}}_t = \tilde{\mathbf{B}}_t^\top \tilde{\mathbf{B}}_t$ is guaranteed with $\mathbf{S}_t = \tilde{\mathbf{S}}_t$, thus enabling a unified step size. With $\eta$ too small however, it is more beneficial (compared to balanced updates) to scale either $\mathbf{A}_t$ or $\mathbf{B}_t$ to accommodate the small $\eta$. This corresponds to the two solutions in the $0 < \eta < 1/(\tilde{C}_t L)$ case of (10).

### 3.3 RefLoRA: Refactored low-rank adaptation

Having identified in Theorem 3 the optimal $\mathbf{S}_t^*$ minimizing the loss upper bound, we are ready to introduce our refactored low-rank adaptation (RefLoRA) approach. RefLoRA substitutes LoRA's per-step update (1) with the refactored version (3), where $\tilde{\mathbf{A}}_t = \mathbf{A}_t \mathbf{P}_t$, $\tilde{\mathbf{B}}_t = \mathbf{B}_t \mathbf{P}_t^{-\top}$, and $\mathbf{P}_t \mathbf{P}_t^\top = \mathbf{S}_t^*$. As the right singular matrix $\mathbf{V}_t^P \in \mathrm{O}(r)$ can be arbitrary (cf. Section 3.1), one convenient choice is to simply set $\mathbf{P}_t = \mathbf{S}_t^{*1/2}$. Next, this subsection deals with two practical challenges facing the implementation of RefLoRA, and delves into several important properties of the resultant approach.

**Smoothness constant** $L$ is typically unknown and difficult to estimate in practice, especially for LLMs. Thus, it is unclear when to switch between the two schemes in (10). As a remedy, one can either treat $1/L$ as a hyperparameter akin to GD, or, adhere to the balanced update by choosing $\mathbf{S}_t = \tilde{\mathbf{S}}_t$ for $\forall\eta$. The latter results in a continuous loss function and upper bounds as sketched in Figure 1b. Since this adjustment only affects the region where $\eta$ is tiny, it still allows for a larger $\eta$ to improve convergence. For simplicity, the balanced update is adopted thereafter.

**Adaptive optimizers** such as Adam [26] and AdamW [40] are default to optimizing large models, which adjust the update using the first moment and entrywise second moment estimated from the running average of stochastic gradients. When refactoring $(\mathbf{A}_t, \mathbf{B}_t)$ to $(\mathbf{A}_t \mathbf{P}_t, \mathbf{B}_t \mathbf{P}_t^{-\top})$, the first moment estimator can be transformed accordingly, while the second moment is generally intractable due to the entrywise square. To tackle this challenge, we "refactor back" the low-rank matrices after the GD update in (3) by right-multiplying $\mathbf{P}_t^{-1}$ and $\mathbf{P}_t^\top$. This gives an alternative update

$$\mathbf{A}_{t+1} := (\tilde{\mathbf{A}}_t + \Delta\tilde{\mathbf{A}}_t)\mathbf{P}_t^{-1} = \mathbf{A}_t - \eta\nabla\ell(\mathbf{W}_t)\mathbf{B}_t \tilde{\mathbf{S}}_t^{-1}, \tag{11a}$$

$$\mathbf{B}_{t+1} := (\tilde{\mathbf{B}}_t + \Delta\tilde{\mathbf{B}}_t)\mathbf{P}_t^\top = \mathbf{B}_t - \eta\nabla\ell(\mathbf{W}_t)^\top \mathbf{A}_t \tilde{\mathbf{S}}_t \tag{11b}$$

whose axes align with the original $(\mathbf{A}_t, \mathbf{B}_t)$, thus waiving the need to transform the moment estimators. This observation prompts the view of refactoring as GD with preconditioning matrices $\tilde{\mathbf{S}}_t^{-1}$ and $\tilde{\mathbf{S}}_t$. As a side benefit, this also eliminates the need to compute $\mathbf{P}_t$ from $\tilde{\mathbf{S}}_t$.

Table 1: Additional complexities introduced by LoRA variants

| Method | Time | Space |
|---|---|---|
| LoRA forward/backward | $\Omega(mn)$ | $\Omega(mn)$ |
| LoRA-Pro [61] | $\mathcal{O}(m^2 r + (m+n+r)r^2)$ | $\mathcal{O}(m^2 + (m+n+r)r)$ |
| LoRA-RITE [67] | $\mathcal{O}((m+n+r)r^2)$ | $\mathcal{O}((m+n+r)r)$ |
| RefLoRA (Thm. 3) | $\mathcal{O}((m+n+r)r^2)$ | $\mathcal{O}(r^2)$ |
| RefLoRA-S (Thm. 5) | $\mathcal{O}((m+n)r)$ | $\mathcal{O}(1)$ |

Interestingly, this reformulation (11) offers an alternative interpretation of RefLoRA as **Riemannian optimization over the quotient manifold** $\mathcal{M}_r \cong \left(\mathbb{R}_*^{m \times r} \times \mathbb{R}_*^{n \times r}\right) / \mathrm{GL}(r)$ [5], where the quotient is wrt equivalence relation $(\mathbf{A}, \mathbf{B}) \sim (\mathbf{AP}, \mathbf{BP}^{-\top})$, $\mathbf{P} \in \mathrm{GL}(r)$, and the subscript $*$ denotes full-rank matrices. On this manifold, (11) can be equivalently derived from the Riemannian metric

$$g_{(\mathbf{A},\mathbf{B})}((\mathbf{G}_A, \mathbf{G}_B), (\mathbf{Z}_A, \mathbf{Z}_B)) := \langle \mathbf{G}_A \tilde{\mathbf{S}}, \mathbf{Z}_A \rangle_F + \langle \mathbf{G}_B \tilde{\mathbf{S}}^{-1}, \mathbf{Z}_B \rangle_F. \tag{12}$$

Denoting (9) as $\tilde{\mathbf{S}}(\mathbf{A}, \mathbf{B})$, it can be verified that the metric (12) is congruence-invariant $\tilde{\mathbf{S}}(\mathbf{AP}, \mathbf{BP}^{-\top}) = \mathbf{P}^{-1}\tilde{\mathbf{S}}(\mathbf{A}, \mathbf{B})\mathbf{P}^{-\top}$. Moreover, it ensures the weight update always lies in the horizontal space altering the equivalence class, and never wastes efforts on the vertical (gauge) directions moving within the same equivalent group; see proof in Appendix B.7. As shown in the next section, this steepest descent on horizontal space brings about a significant faster convergence.

Next, we present three key RefLoRA properties.

**Balanced refactoring** $\mathbf{A}_t^\top \mathbf{A}_t = \mathbf{B}_t^\top \mathbf{B}_t$ can be achieved per iteration upon setting $\mathbf{P}_t \mathbf{P}_t^\top = \tilde{\mathbf{S}}_t$, as stated in Section 3.2. It is worthwhile pointing out that SVD-based initializations including PiSSA [41] and LoftQ [34] inherently satisfy $\mathbf{A}_0^\top \mathbf{A}_0 = \mathbf{B}_0^\top \mathbf{B}_0^\top$. When $t$ is small, the balance tends to hold approximately even without refactoring, which partially explains why empirical convergence is fast during the early epochs. Analytically, balanced refactoring maximizes the potential loss reduction as formalized in the ensuing Theorem.

**Theorem 4.** *The solution* $\mathbf{S}_t = \tilde{\mathbf{S}}_t$ *given by* (9) *minimizes the lower bound*

$$0 \geq \underbrace{\langle \nabla_{\tilde{\mathbf{A}}_t} \ell(\tilde{\mathbf{W}}_t), \Delta\tilde{\mathbf{A}}_t \rangle_F + \langle \nabla_{\tilde{\mathbf{B}}_t} \ell(\tilde{\mathbf{W}}_t), \Delta\tilde{\mathbf{B}}_t \rangle_F}_{\approx \Delta\ell(\mathbf{W}_t) := \ell(\mathbf{W}_{t+1}) - \ell(\mathbf{W}_t) \text{ with small } \eta} \geq -\eta\|\nabla\ell(\mathbf{W}_t)\|_2^2 \big(\|\mathbf{A}_t \mathbf{S}_t^{\frac{1}{2}}\|_F^2 + \|\mathbf{B}_t \mathbf{S}_t^{-\frac{1}{2}}\|_F^2\big).$$

The upper bound 0 stems from the low-rank nature of LoRA, which can be reached when $\nabla\ell(\mathbf{W}_t)^\top \in \mathrm{Null}(\mathbf{B}_t^\top)$ and $\nabla\ell(\mathbf{W}_t) \in \mathrm{Null}(\mathbf{A}_t^\top)$; i.e., the stationary points of $\mathbf{A}_t$ and $\mathbf{B}_t$. While this upper bound cannot be improved, RefLoRA minimizes the lower bound, and thus leads to a more effective descent in the loss.

**Additional computational overhead** induced by RefLoRA is as small as $\mathcal{O}((m+n+r)r^2)$ in time and $\mathcal{O}(r^2)$ in memory, thanks to the decoupling of $\nabla\ell(\mathbf{W}_t)$ in Proposition (2). Compared to the forward/backward overhead $\Omega(mn)$ of LoRA, the extra complexity introduced by RefLoRA is relatively minimal. In contrast, LoRA-Pro [61] suffers from $\mathcal{O}(m^2 r + (m+n+r)r^2)$ time and $\mathcal{O}(m^2 + (m+n+r)r)$ space, whereas LoRA-RITE [67] requires $\mathcal{O}((m+n+r)r)$ memory for polar decomposition. Despite the low complexity of RefLoRA, further curtailing the extra cost can be beneficial for resource-limited applications. The next theorem restricts $\mathbf{S}_t$ to a scaled identity $s_t \mathbf{I}_r$ with $s_t > 0$, and derives a result analogous to Theorem 3.

**Theorem 5.** *Consider* (8) *confined with* $\mathbf{S}_t = s_t \mathbf{I}_r$, $s_t \in \mathbb{R}_{++}$. *Under Assumptions 1 and 2, it holds*

$$s_t^* = \begin{cases} \dfrac{\|\mathbf{B}_t\|_F}{\|\mathbf{A}_t\|_F}, & \text{if } \eta \geq \dfrac{1}{2\|\mathbf{A}_t\|_F\|\mathbf{B}_t\|_F L} \text{ or } \eta < 0 \\ \dfrac{\frac{1}{L\eta} \pm \sqrt{\frac{1}{L^2\eta^2} - 4\|\mathbf{A}_t\|_F^2\|\mathbf{B}_t\|_F^2}}{2\|\mathbf{A}_t\|_F^2}, & \text{if } 0 < \eta < \dfrac{1}{2\|\mathbf{A}_t\|_F\|\mathbf{B}_t\|_F L} \end{cases}. \tag{13}$$

This simplified refactoring (RefLoRA-S) enjoys further reduced complexity of $\mathcal{O}((m+n)r)$ time and $\mathcal{O}(1)$ space; see Table 1 for a summary. Compared to Theorem 3, the scalar $s_t$ only has two global optima when $\eta$ is too small. Note that this simplification only guarantees a weaker balance $\|\tilde{\mathbf{A}}_t\|_F = \|\tilde{\mathbf{B}}_t\|_F$. However, it can be directly combined with adaptive optimizers by scaling the moment estimators without the second refactoring.

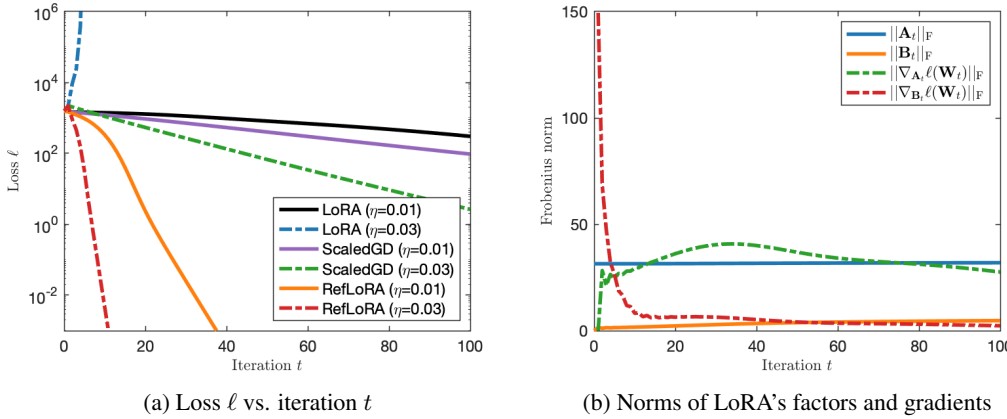

(a) Loss $\ell$ vs. iteration $t$         (b) Norms of LoRA's factors and gradients

Figure 2: Comparison of LoRA, ScaledGD, and RefLoRA for matrix factorization

**Consistent weight updates** are always guaranteed for all equivalent factorizations.

**Theorem 6.** *Under Assumptions 1-2 and for any $\mathbf{A}_t' \mathbf{B}_t'^{\top} = \mathbf{A}_t \mathbf{B}_t^{\top}$, let $\Delta \tilde{\mathbf{W}}_t'$ and $\Delta \tilde{\mathbf{W}}_t$ be the corresponding weight updates* (5) *with RefLoRA. It then always holds that $\Delta \tilde{\mathbf{W}}_t' = \Delta \tilde{\mathbf{W}}_t$.*

This consistency holds for either $\mathbf{S}_t = \mathbf{S}_t^*$ or $\mathbf{S}_t = \tilde{\mathbf{S}}_t$, but not for the lightweight version (13). In this context, we dealt with slow convergence and the non-uniqueness of low-rank factorization with the added benefit of balanced updates. The step-by-step algorithm of RefLoRA(-S) is listed in Appendix C. Next, experiments are conducted to verify our findings.

## 4 Numerical tests

This section evaluates the empirical performance of RefLoRA. All experimental setups including platforms, datasets, models, metrics, and hyperparameters are detailed in Appendix D. Our implementation is available at `https://github.com/zhangyilang/RefLoRA`.

### 4.1 Matrix factorization

The first numerical test considers low-rank matrix factorization $\min_{\mathbf{A},\mathbf{B}} \frac{1}{2} \|\mathbf{Y} - \mathbf{AB}\|_{\mathrm{F}}^2$, where $\mathbf{Y} \in \mathbb{R}^{m \times n}$ is a given low-rank matrix [14]. It can be viewed as applying LoRA to a single-layer model, and training it with whitened data [2, 31]. Figure 2a compares the training loss of RefLoRA with LoRA, and another popular approach dubbed ScaledGD [56]—which is tailored particularly for low-rank matrix factorization, and has gained popularity among LoRA variants; see e.g., [69]. Each method is tested with two learning rates $\eta \in \{0.01, 0.03\}$. While vanilla LoRA converges slowly with the lower learning rate and diverges with the higher one, RefLoRA remains stable and converges markedly faster under both rates, as corroborated also with observations in Figure 1. Notably, RefLoRA even outperforms ScaledGD, thanks to its balanced update. Figure 2b depicts the dynamics of LoRA ($\eta = 0.01$) by plotting the Frobenius norms of $\mathbf{A}_t, \mathbf{B}_t$ and their gradients. It is observed that $\|\mathbf{A}_t\|_{\mathrm{F}}$ and $\|\mathbf{B}_t\|_{\mathrm{F}}$ are highly unbalanced across iterations, and $\|\nabla_{\mathbf{A}_t} \ell\|_{\mathrm{F}}$ as well as $\|\nabla_{\mathbf{B}_t} \ell\|_{\mathrm{F}}$ exhibit a sharp change when $t$ is small. In comparison, RefLoRA maintains $\tilde{\mathbf{A}}_t^{\top} \tilde{\mathbf{A}}_t = \tilde{\mathbf{B}}_t^{\top} \tilde{\mathbf{B}}_t, \forall t$, which guarantees that $\|\tilde{\mathbf{A}}_t\|_{\mathrm{F}}^2 = \mathrm{tr}(\tilde{\mathbf{A}}_t^{\top} \tilde{\mathbf{A}}_t) = \mathrm{tr}(\tilde{\mathbf{B}}_t^{\top} \tilde{\mathbf{B}}_t) = \|\tilde{\mathbf{B}}_t\|_{\mathrm{F}}$. This balance can afford larger learning rates, thus improving the empirical convergence.

### 4.2 Natural language understanding

Beyond matrix factorization the evaluation here starts with fine-tuning DeBERTaV3-base [21], a masked language model with 184M parameters, on the General Language Understanding Evaluation (GLUE) benchmark [59]. GLUE contains 8 datasets, providing a general-purpose evaluation for natural language understanding (NLU) [59]. The test setup follows from [23, 70], where LoRA rank is $r = 8$, reducing the trainable parameters to 1.33M. We compare RefLoRA and its lightweight variant RefLoRA-S against a suite of LoRA variants, including SOTA methods DoRA [65] and

Table 2: Performance comparison using DeBERTaV3-base on the GLUE benchmark dataset. The best results are depicted in solid lines. The score in the last column averages Matthews correlation coefficient (Mcc), accuracies (Acc), Pearson correlation (Corr), and matched accuracy (M). The results are obtained by averaging 5 random runs.

| Method | Params | CoLA | SST-2 | MRPC | STS-B | QQP | MNLI | QNLI | RTE | All |
|---|---|---|---|---|---|---|---|---|---|---|
| | | Mcc | Acc | Acc | Corr | Acc/F1 | M/Mm | Acc | Acc | Avg |
| Full FT | 184M | 69.19 | 95.63 | 89.46 | 91.60 | 92.40/89.80 | 89.90/90.12 | 94.03 | 83.75 | 88.25 |
| BitFit | 0.1M | 66.96 | 94.84 | 87.75 | 91.35 | 88.41/84.95 | 89.37/89.91 | 92.24 | 78.70 | 86.20 |
| HAdapter | 1.22M | 68.64 | 95.53 | 89.95 | 91.48 | 91.91/89.27 | 90.13/90.17 | 94.11 | 84.48 | 88.28 |
| PAdapter | 1.18M | 68.77 | 95.61 | 89.46 | 91.54 | 92.04/89.40 | 90.33/90.39 | 94.29 | 85.20 | 88.41 |
| LoRA | 1.33M | 69.82 | 94.95 | 89.95 | 91.60 | 91.99/89.38 | 90.65/90.69 | 93.87 | 85.20 | 88.50 |
| DoRA | 1.33M | 70.85 | 95.79 | 90.93 | 91.79 | 92.07/- | 90.29/- | 94.10 | 86.04 | 88.98 |
| AdaLoRA | 1.27M | 71.45 | **96.10** | 90.69 | 91.84 | 92.23/89.74 | **90.76/90.79** | **94.55** | 88.09 | 89.46 |
| LoRA-Pro | 1.33M | 71.36 | 95.76 | 90.20 | 91.92 | 92.19/89.60 | 90.23/90.19 | 94.29 | 85.56 | 88.94 |
| LoRA-RITE | 1.33M | 69.55 | 95.41 | 90.93 | 91.79 | 92.02/89.42 | 90.22/90.33 | 94.42 | 85.20 | 88.69 |
| RefLoRA | 1.33M | **71.73** | 95.99 | **91.42** | 92.03 | 92.28/89.70 | 90.23/90.41 | 94.40 | **88.09** | **89.52** |
| RefLoRA-S | 1.33M | 70.66 | 95.76 | 90.44 | **92.21** | **92.43/89.89** | 90.13/90.17 | 94.16 | 87.73 | 89.19 |

AdaLoRA [70], as well as baselines falling in the same category with RefLoRA, i.e., LoRA-Pro [61] and LoRA-RITE [67]. The results can be found in Table 2. It is worth mentioning that these datasets are relatively small, so that full fine-tuning (FT) is prone to overfitting, thus leading to worse performance compared to PEFT methods. RefLoRA and RefLoRA-S outperform all competitors on 5 out of 8 datasets, and present comparable performance to SOTA approaches on the rest 3 datasets. Overall, RefLoRA achieves the highest average performance, demonstrating more effective optimization via refactoring. In spite of the simplified refactoring, RefLoRA-S maintains competitive performance, i.e., only $0.33\%$ lower than RefLoRA on average, while markedly reducing computational overhead.

## 4.3  Commonsense reasoning

We further extend our numerical experiments to fine-tuning the LLaMA series [57, 58, 17], which are autoregressive language models with 7B and 8B parameters. We tackle commonsense reasoning tasks following the setup in [24, 65]. Training data are aggregated from 8 datasets listed in Table 3, and test sets remain separate for individual evaluation. These reasoning tasks are intended to push the model beyond pattern recognition, requiring commonsense and knowledge to make proper inferences. The baselines are chosen as DoRA [65], LoRA-RITE [67], PrecLoRA [69], and NoRA+ [31]. Note that the latter two approaches are variants of ScaledGD [56], sharing similar complexity with RefLoRA. The accuracy comparison is summarized in Table 3. Both RefLoRA and RefLoRA-S consistently outperform other PEFT methods in 5 out of 6 settings. Even under lower-rank configurations $r = 16$, both RefLoRA and RefLoRA-S continue to lead or match top-performing approaches, underscoring their parameter efficiency and robustness. These results demonstrate the effectiveness of the proposed RefLoRA(-S), and underscore the potential of optimal refactoring.

## 4.4  Subject-driven image generation with diffusion models

Akin to LoRA, RefLoRA can be seamlessly integrated into a wide range of larger models beyond LLMs. Further numerical tests are conducted on a subject-driven image generation task [48] with Stable Diffusion v1.4 [45]. The goal is to fine-tune a diffusion model using a few user-provided images so that it can generate the same object in various contexts. Specifically, the model is fine-tuned on a set of images labeled "a photo of sks dog," and subsequently evaluated by generating images under the prompt "a sks dog eating nachos." LoRA adapters with rank $r = 4$ are attached to the U-Net component of Stable Diffusion, and experimental setups including hyperparameters are default to those in [48]. The fine-tuning losses for LoRA, LoRA-Pro, LoRA-RITE, and RefLoRA are summarized in the table 4, where average and standard deviation are calculated over 3 random runs. It is seen that RefLoRA achieves $14.0\%$, $13.1\%$, and $9.5\%$ improvements over LoRA, LoRA-Pro, and LoRA-RITE, respectively. In addition to quantitative gains in loss, we also observe noticeable improvements in image quality; see Figure 3. The generations from RefLoRA-tuned models show

Table 3: Accuracy comparison using LLaMA series on commonsense reasoning datasets.

| | r | Method | Params | BoolQ | PIQA | SIQA | HS | WG | ARCe | ARCc | OBQA | Avg |
|---|---|---|---|---|---|---|---|---|---|---|---|---|
| | | ChatGPT-3.5-turbo | - | 73.1 | 85.4 | 68.5 | 78.5 | 66.1 | 89.8 | 79.9 | 74.8 | 77.0 |
| LLaMA-7B | 32 | LoRA | 0.83% | 66.42 | 80.03 | 77.84 | 82.88 | 81.85 | 79.92 | 63.40 | 77.20 | 76.19 |
| | | PrecLoRA | 0.83% | 68.96 | 80.95 | 77.43 | 81.54 | 80.27 | 78.83 | 64.16 | 79.20 | 76.42 |
| | | NoRA+ | 0.83% | 69.85 | 81.83 | 77.38 | 82.09 | 80.03 | 79.67 | 64.25 | 78.60 | 76.71 |
| | | DoRA | 0.84% | 69.7 | 83.4 | 78.6 | 87.2 | 81.0 | 81.9 | 66.2 | 79.2 | 78.4 |
| | | LoRA-RITE | 0.84% | 69.82 | 82.75 | 78.55 | 84.72 | 81.69 | 82.15 | 66.23 | 81.40 | 78.54 |
| | | RefLoRA | 0.83% | 69.60 | 82.48 | 79.53 | 88.25 | 82.56 | 81.57 | 66.64 | 80.20 | **78.85** |
| | | RefLoRA-S | 0.83% | 70.18 | 82.48 | 78.15 | 87.41 | 82.08 | 81.52 | 65.36 | 81.60 | 78.60 |
| | 16 | DoRA | 0.43% | 70.0 | 82.6 | 79.7 | 83.2 | 80.6 | 80.6 | 65.4 | 77.6 | 77.5 |
| | | RefLoRA | 0.41% | 69.66 | 82.43 | 79.43 | 87.38 | 81.22 | 80.68 | 65.44 | 78.60 | **78.11** |
| | | RefLoRA-S | 0.41% | 67.65 | 81.50 | 79.07 | 88.28 | 81.77 | 81.23 | 64.59 | 78.60 | 77.84 |
| LLaMA2-7B | 32 | LoRA | 0.83% | 69.8 | 79.9 | 79.5 | 83.6 | 82.6 | 79.8 | 64.7 | 81.0 | 77.6 |
| | | PrecLoRA | 0.83% | 71.47 | 81.50 | 78.81 | 85.97 | 80.43 | 81.14 | 66.55 | 81.00 | 78.36 |
| | | NoRA+ | 0.83% | 70.52 | 81.94 | 79.07 | 87.66 | 82.24 | 82.70 | 67.06 | 80.20 | 78.92 |
| | | DoRA | 0.84% | 71.8 | 83.7 | 76.0 | 89.1 | 82.6 | 83.7 | 68.2 | 82.4 | 79.7 |
| | | LoRA-RITE | 0.84% | 71.04 | 82.43 | 79.79 | 89.12 | 84.53 | 83.88 | 68.77 | 81.20 | 80.10 |
| | | RefLoRA | 0.83% | 72.54 | 83.79 | 80.04 | 86.94 | 84.85 | 86.36 | 71.50 | 80.20 | **80.78** |
| | | RefLoRA-S | 0.83% | 73.36 | 83.84 | 80.76 | 90.02 | 82.48 | 84.55 | 67.92 | 82.60 | 80.69 |
| | 16 | DoRA | 0.43% | 72.0 | 83.1 | 79.9 | 89.1 | 83.0 | 84.5 | 71.0 | 81.2 | **80.5** |
| | | RefLoRA | 0.41% | 71.38 | 82.43 | 80.35 | 90.49 | 83.43 | 84.05 | 69.28 | 82.00 | 80.43 |
| | | RefLoRA-S | 0.41% | 72.08 | 83.03 | 80.45 | 85.89 | 83.27 | 84.30 | 69.88 | 82.00 | 80.11 |
| LLaMA3-8B | 32 | LoRA | 0.70% | 70.8 | 85.2 | 79.9 | 91.7 | 84.3 | 84.2 | 71.2 | 79.0 | 80.8 |
| | | PrecLoRA | 0.70% | 70.73 | 85.80 | 78.86 | 91.87 | 83.66 | 85.10 | 71.08 | 82.40 | 81.19 |
| | | NoRA+ | 0.70% | 71.16 | 85.10 | 79.48 | 92.22 | 83.35 | 85.86 | 72.27 | 83.20 | 81.58 |
| | | DoRA | 0.71% | 74.6 | 89.3 | 79.9 | 95.5 | 85.6 | 90.5 | 80.4 | 85.8 | 85.2 |
| | | LoRA-RITE | 0.84% | 74.19 | 89.44 | 81.52 | 95.44 | 86.74 | 90.45 | 80.12 | 86.60 | 85.56 |
| | | RefLoRA | 0.70% | 75.35 | 88.74 | 80.91 | 95.71 | 86.66 | 90.49 | 80.20 | 87.40 | 85.68 |
| | | RefLoRA-S | 0.70% | 75.50 | 89.72 | 81.11 | 95.59 | 87.29 | 90.99 | 79.78 | 86.00 | **85.75** |
| | 16 | DoRA | 0.35% | 74.5 | 88.8 | 80.3 | 95.5 | 84.7 | 90.1 | 79.1 | 87.2 | 85.0 |
| | | RefLoRA | 0.35% | 75.26 | 88.79 | 81.37 | 95.85 | 85.64 | 90.11 | 80.55 | 86.60 | **85.52** |
| | | RefLoRA-S | 0.35% | 74.92 | 89.01 | 80.60 | 95.75 | 85.24 | 90.45 | 80.89 | 86.40 | 85.41 |

Table 4: Fine-tuning loss (↓) for subject-driven image generation on DreamBooth.

| Loss | LoRA | LoRA-Pro | LoRA-RITE | RefLoRA |
|---|---|---|---|---|
| Avg±std | $0.100 \pm 0.015$ | $0.099 \pm 0.015$ | $0.095 \pm 0.016$ | $\mathbf{0.086} \pm 0.017$ |

markedly clearer details and better object fidelity, particularly in the mouth and tongue regions, where features are often distorted in outputs from other three baselines.

## 4.5 Convergence and complexity comparison

Lastly, we evaluate the convergence behavior and computational efficiency of RefLoRA in comparison with LoRA [23], LoRA-Pro [61], and LoRA-RITE [67]. These tests are conducted on the MRPC subset of the GLUE benchmark using DeBERTaV3-base as the backbone model. For fairness, the learning rate is set to $\eta = 4 \times 10^{-4}$ across all methods. Figure 4a depicts the loss $\ell(\mathbf{W}_t)$ over 10 fine-tuning epochs. The loss of RefLoRA(-S) declines more rapidly and exhibits less fluctuation than the other three methods, which ultimately achieves the lowest value approaching 0. This confirms stability and convergence speed, on par with our theoretical insights in Section 3. The sharper descent in the loss suggests improved optimization trajectories enabled by RefLoRA's principled refactoring.

The comparison of computational overhead is presented in Figure 4b. Time complexity is reflected via the fine-tuning throughput (iterations per second; higher is better), while space complexity is measured in GPU memory occupation (lower is better). For better visualization, the vertical axes start with a non-zero value. The plot reveals that RefLoRA and RefLoRA-S respectively showcase

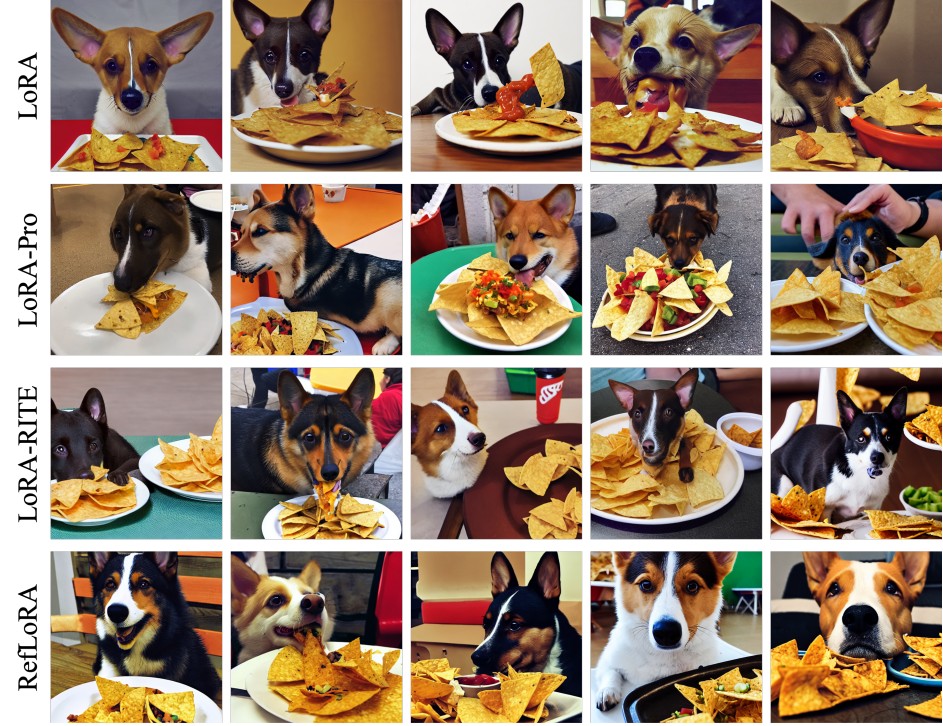

Figure 3: Images generated from Stable Diffusion fine-tuned with different approaches.

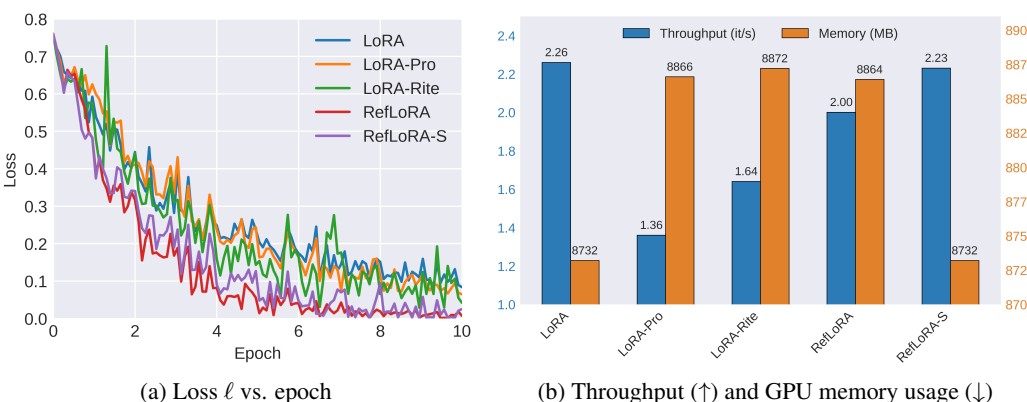

(a) Loss $\ell$ vs. epoch

(b) Throughput ($\uparrow$) and GPU memory usage ($\downarrow$)

Figure 4: Convergence and complexity comparison

88.5% and 98.7% throughput compared to LoRA, at the additional memory cost of 132MB and $< 1$MB. In contrast, the throughput of LoRA-Pro and LoRA-RITE are 60.2% and 72.6% of LoRA, requiring 134MB and 140MB extra memory. This is consistent with our complexity analysis in Table 1. Extended comparisons scaling up to 27B models are offered in Appendix E.

## 5 Conclusion and outlook

This paper introduced refactored low-rank adaptation (RefLoRA), a principled LoRA variant that remarkably enhances the efficiency and stability of fine-tuning large models. By identifying the optimal matrix $\mathbf{S}_t$ that minimizes the loss upper bound, RefLoRA addressed the key challenges in LoRA by providing balanced updates, improved convergence, and consistently superior empirical performance with affordable overhead. To further facilitate scalability and applicability to large models, RefLoRA-S leverages simplified refactoring to minimize the computational complexity. Our future research agenda involves analyzing the convergence rate of LoRA and RefLoRA, and adapting RefLoRA to extensive model architectures such as vision transformers and diffusion models.

## Acknowledgments

This work is partly supported by NSF grants 2220292, 2212318, and 2312547. B. Li is supported by Swiss National Science Foundation (SNSF) Project Funding No. 200021-207343.

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

# A   Additional related work

**LoRA variants.** LoRA has been extended in several directions. For example, recent works further reduce LoRA's trainable parameters [28, 37, 15, 19, 27]. As these methods can achieve performance comparable to that of LoRA, they indirectly suggest that the expressiveness of LoRA has not been fully exploited. Another line of work [12, 34] incorporates quantization in LoRA to reduce memory footprint and computational overhead. There are also works that broaden the applicability of LoRA to pre-training LLMs by e.g., sequentially chaining LoRA modules [35, 39]. As our method enhances the efficiency of LoRA training, we expect that it can be seamlessly integrated to such settings as well. Going beyond LoRA, there are also approaches for fine-tuning LLMs in a parameter efficient manner such as [33, 29, 38, 64]. Moreover, [30] shows that applying sharpness-aware minimization on LoRA promotes balance between $\mathbf{A}_t$ and $\mathbf{B}_t$. These approaches are orthogonal to our work.

**Broader Impact.** The theoretical insights and algorithmic contributions of the current work are broadly applicable across a range of fine-tuning scenarios. Our RefLoRA enhances the efficiency and effectiveness of adapting language models to downstream tasks, leading to improved performance in applications such as sentiment classification. This, in turn, can positively impact real-world systems including recommendation systems by increasing accuracy and relevance. However, caution should be exercised when deploying the method for generative tasks. In such settings, the outputs of language models should be carefully reviewed, and proper safeguards, such as gating mechanisms, should be considered to ensure safety, reliability, and trustworthiness of the generated content.

**Future directions.** Due to limited computational resources, our evaluation currently deals with models having reasonably large scale, e.g., LLaMA3-8B. Our future work will include scaling RefLoRA to even larger models, such as those with 30B parameters. Another promising direction is to integrate RefLoRA with sequentially chaining, namely [35]. This will further broaden the applicability of RefLoRA for pre-training LLMs.

# B   Missing proofs

This appendix presents the proofs that were omitted from the main paper.

## B.1   Proof of Lemma 1

*Proof.* First, we prove (4) by showing that the two sets in (4) contain each other.

For any pair $(\mathbf{A}_t \mathbf{P}_t, \mathbf{B}_t \mathbf{P}_t^{-\top})$, it is easy to see $\mathbf{A}_t \mathbf{P}_t (\mathbf{B}_t \mathbf{P}_t^{-\top})^\top = \mathbf{A}_t \mathbf{B}_t^\top$. Thus we have

$$\{(\tilde{\mathbf{A}}_t, \tilde{\mathbf{B}}_t) \mid \tilde{\mathbf{A}}_t \tilde{\mathbf{B}}_t^\top = \mathbf{A}_t \mathbf{B}_t^\top\} \supseteq \{(\mathbf{A}_t \mathbf{P}_t, \mathbf{B}_t \mathbf{P}_t^{-\top}) \mid \mathbf{P}_t \in \mathrm{GL}(r)\}.$$

Next, we prove the opposite containing relationship. Let $(\tilde{\mathbf{A}}_t, \tilde{\mathbf{B}}_t)$ be an arbitrary pair satisfying $\tilde{\mathbf{A}}_t \tilde{\mathbf{B}}_t^\top = \mathbf{A}_t \mathbf{B}_t^\top$. It follows that

$$\mathrm{rank}(\tilde{\mathbf{A}}_t \tilde{\mathbf{B}}_t^\top) = \mathrm{rank}(\mathbf{A}_t \mathbf{B}_t^\top) \geq \mathrm{rank}(\mathbf{A}_t) + \mathrm{rank}(\mathbf{B}_t) - r = r,$$
$$\mathrm{rank}(\tilde{\mathbf{A}}_t \tilde{\mathbf{B}}_t^\top) \leq \min\{\mathrm{rank}(\tilde{\mathbf{A}}_t), \mathrm{rank}(\tilde{\mathbf{B}}_t)\} \leq r.$$

We thus obtain $\mathrm{rank}(\mathbf{A}_t \mathbf{B}_t^\top) = \mathrm{rank}(\tilde{\mathbf{A}}_t \tilde{\mathbf{B}}_t^\top) = r$, and $\mathrm{rank}(\tilde{\mathbf{A}}_t) = \mathrm{rank}(\tilde{\mathbf{B}}_t) = r$.

Since $\mathrm{Col}(\mathbf{A}_t \mathbf{B}_t^\top) \subseteq \mathrm{Col}(\mathbf{A}_t)$ and $\dim(\mathrm{Col}(\mathbf{A}_t \mathbf{B}_t^\top)) = \mathrm{rank}(\mathbf{A}_t \mathbf{B}_t^\top) = r = \mathrm{rank}(\mathbf{A}_t) = \dim(\mathrm{Col}(\mathbf{A}_t))$, we have $\mathrm{Col}(\mathbf{A}_t \mathbf{B}_t^\top) = \mathrm{Col}(\mathbf{A}_t)$; and likewise $\mathrm{Col}(\tilde{\mathbf{A}}_t \tilde{\mathbf{B}}_t^\top) = \mathrm{Col}(\tilde{\mathbf{A}}_t)$.

Then, the condition $\tilde{\mathbf{A}}_t \tilde{\mathbf{B}}_t^\top = \mathbf{A}_t \mathbf{B}_t^\top$ leads to

$$\mathrm{Col}(\tilde{\mathbf{A}}_t) = \mathrm{Col}(\tilde{\mathbf{A}}_t \tilde{\mathbf{B}}_t^\top) = \mathrm{Col}(\mathbf{A}_t \mathbf{B}_t^\top) = \mathrm{Col}(\mathbf{A}_t).$$

This suggests, there must exist an invertible matrix $\mathbf{P}_t \in \mathbb{R}^{r \times r}$ such that $\tilde{\mathbf{A}}_t = \mathbf{A}_t \mathbf{P}_t$.

As a result, we have $\mathbf{A}_t \mathbf{P}_t \tilde{\mathbf{B}}_t^\top = \mathbf{A}_t \mathbf{B}_t^\top$. Multiplying $\mathbf{P}_t^{-1} \mathbf{A}_t^\dagger$ on both sides and taking transpose yield $\tilde{\mathbf{B}}_t = \mathbf{B}_t \mathbf{P}_t^{-\top}$. This suggests

$$\{(\tilde{\mathbf{A}}_t, \tilde{\mathbf{B}}_t) \mid \tilde{\mathbf{A}}_t \tilde{\mathbf{B}}_t^\top = \mathbf{A}_t \mathbf{B}_t^\top\} \subseteq \{(\mathbf{A}_t \mathbf{P}_t, \mathbf{B}_t \mathbf{P}_t^{-\top}) \mid \mathbf{P}_t \in \mathrm{GL}(r)\}.$$

which completes the proof of (4).

Additionally, it follows from the definition of $\Delta\tilde{\mathbf{W}}_t$ that

$$
\begin{aligned}
\Delta\tilde{\mathbf{W}}_t &= \tilde{\mathbf{A}}_t\Delta\tilde{\mathbf{B}}_t^\top + \Delta\tilde{\mathbf{A}}_t\tilde{\mathbf{B}}_t^\top + \Delta\tilde{\mathbf{A}}_t\Delta\tilde{\mathbf{B}}_t^\top \\
&\overset{(a)}{=} -\eta\tilde{\mathbf{A}}_t\tilde{\mathbf{A}}_t^\top\nabla\ell(\mathbf{W}_t) - \eta\nabla\ell(\mathbf{W}_t)\tilde{\mathbf{B}}_t\tilde{\mathbf{B}}_t^\top + \eta^2\nabla\ell(\mathbf{W}_t)\tilde{\mathbf{B}}_t\tilde{\mathbf{A}}_t^\top\nabla\ell(\mathbf{W}_t) \\
&\overset{(b)}{=} -\eta\mathbf{A}_t\mathbf{P}_t\mathbf{P}_t^\top\mathbf{A}_t^\top\nabla\ell(\mathbf{W}_t) - \eta\nabla\ell(\mathbf{W}_t)\mathbf{B}_t\mathbf{P}_t^{-\top}\mathbf{P}_t^{-1}\mathbf{B}_t^\top + \eta^2\nabla\ell(\mathbf{W}_t)\mathbf{B}_t\mathbf{A}_t^\top\nabla\ell(\mathbf{W}_t) \\
&\overset{(c)}{=} \mathbf{A}_t(\mathbf{P}_t\mathbf{P}_t^\top)\Delta\mathbf{B}_t^\top + \Delta\mathbf{A}_t(\mathbf{P}_t\mathbf{P}_t^\top)^{-1}\mathbf{B}_t^\top + \Delta\mathbf{A}_t\Delta\mathbf{B}_t^\top
\end{aligned}
\tag{14}
$$

where $(a)$ uses (3) and that $\mathbf{W}_t^{\mathrm{pt}} + \tilde{\mathbf{A}}_t\tilde{\mathbf{B}}_t^\top = \mathbf{W}_t^{\mathrm{pt}} + \mathbf{A}_t\mathbf{B}_t^\top = \mathbf{W}_t$, $(b)$ is due to $\tilde{\mathbf{A}}_t = \mathbf{A}_t\mathbf{P}_t$ and $\tilde{\mathbf{B}}_t = \mathbf{B}_t\mathbf{P}^{-\top}$, and $(c)$ leverages (1).

Comparing (14) with (2), it can be easily seen that $\mathbf{P}_t\mathbf{P}_t^\top = \mathbf{I}_r$ results in $\Delta\mathbf{W}_t = \Delta\tilde{\mathbf{W}}_t$. $\qquad\square$

## B.2 Proof of Proposition 2

*Proof.* For (3), it follows from Assumption 2 that

$$
\begin{aligned}
\ell(\mathbf{W}_t + \Delta\tilde{\mathbf{W}}_t(\mathbf{S}_t)) &\leq \ell(\mathbf{W}_t) + \langle\nabla\ell(\mathbf{W}_t), \Delta\tilde{\mathbf{W}}_t(\mathbf{S}_t)\rangle_{\mathrm{F}} + \frac{L}{2}\|\Delta\tilde{\mathbf{W}}_t(\mathbf{S}_t)\|_{\mathrm{F}}^2 \\
&\overset{(a)}{=} \ell(\mathbf{W}_t) + \langle\nabla\ell(\mathbf{W}_t), \mathbf{A}_t\mathbf{S}_t\Delta\mathbf{B}_t^\top + \Delta\mathbf{A}_t\mathbf{S}_t^{-1}\mathbf{B}_t^\top - \Delta\mathbf{A}_t\Delta\mathbf{B}_t^\top\rangle_{\mathrm{F}} + \\
&\quad \frac{L}{2}\|\mathbf{A}_t\mathbf{S}_t\Delta\mathbf{B}_t^\top + \Delta\mathbf{A}_t\mathbf{S}_t^{-1}\mathbf{B}_t^\top - \Delta\mathbf{A}_t\Delta\mathbf{B}_t^\top\|_{\mathrm{F}}^2 \\
&\overset{(b)}{=} \langle\nabla\ell(\mathbf{W}_t), \mathbf{A}_t\mathbf{S}_t\Delta\mathbf{B}_t^\top + \Delta\mathbf{A}_t\mathbf{S}_t^{-1}\mathbf{B}_t^\top\rangle_{\mathrm{F}} + \frac{L}{2}\Big[\|\mathbf{A}_t\mathbf{S}_t\Delta\mathbf{B}_t^\top\|_{\mathrm{F}}^2 + \|\Delta\mathbf{A}_t\mathbf{S}_t^{-1}\mathbf{B}_t^\top\|_{\mathrm{F}}^2 + \\
&\quad 2\langle\mathbf{A}_t\mathbf{S}_t\Delta\mathbf{B}_t^\top, \Delta\mathbf{A}_t\mathbf{S}_t^{-1}\mathbf{B}_t^\top\rangle_{\mathrm{F}} - 2\langle\mathbf{A}_t\mathbf{S}_t\Delta\mathbf{B}_t^\top + \Delta\mathbf{A}_t\mathbf{S}_t^{-1}\mathbf{B}_t^\top, \Delta\mathbf{A}_t\Delta\mathbf{B}_t^\top\rangle_{\mathrm{F}}\Big] + \mathrm{Const.} \\
&\overset{(c)}{=} \frac{L}{2}\Big[\Big\|\frac{1}{L}\nabla\ell(\mathbf{W}_t) + \mathbf{A}_t\mathbf{S}_t\Delta\mathbf{B}_t^\top\Big\|_{\mathrm{F}}^2 + \Big\|\frac{1}{L}\nabla\ell(\mathbf{W}_t) + \Delta\mathbf{A}_t\mathbf{S}_t^{-1}\mathbf{B}_t^\top\Big\|_{\mathrm{F}}^2 + \\
&\quad 2\langle\mathbf{A}_t\mathbf{S}_t\Delta\mathbf{B}_t^\top, \Delta\mathbf{A}_t\mathbf{S}_t^{-1}\mathbf{B}_t^\top\rangle_{\mathrm{F}} - 2\langle\mathbf{A}_t\mathbf{S}_t\Delta\mathbf{B}_t^\top + \Delta\mathbf{A}_t\mathbf{S}_t^{-1}\mathbf{B}_t^\top, \Delta\mathbf{A}_t\Delta\mathbf{B}_t^\top\rangle_{\mathrm{F}}\Big] + \mathrm{Const.}
\end{aligned}
\tag{15}
$$

where $(a)$ relies on (14); $(b)$ expands the squared Frobenius norm and merges terms independent $\mathbf{S}_t$ of into $\mathrm{Const.}$; and $(c)$ utilizes completing the square and merges the constant terms.

Next, we bound the four terms in (15). Using the definition (1) of $\Delta\mathbf{B}_t$, the first term is relaxed via

$$
\begin{aligned}
\Big\|\frac{1}{L}\nabla\ell(\mathbf{W}_t) + \mathbf{A}_t\mathbf{S}_t\Delta\mathbf{B}_t^\top\Big\|_{\mathrm{F}}^2 &= \Big\|\frac{1}{L}\nabla\ell(\mathbf{W}_t) - \eta\mathbf{A}_t\mathbf{S}_t\mathbf{A}_t^\top\nabla\ell(\mathbf{W}_t)\Big\|_{\mathrm{F}}^2 \\
&\leq \eta^2\|\nabla\ell(\mathbf{W}_t)\|_2^2\Big\|\frac{1}{L\eta}\mathbf{I}_m - \mathbf{A}_t\mathbf{S}_t\mathbf{A}_t^\top\Big\|_{\mathrm{F}}^2.
\end{aligned}
\tag{16}
$$

Likewise, the second term is bounded through

$$
\Big\|\frac{1}{L}\nabla\ell(\mathbf{W}_t) + \Delta\mathbf{A}_t\mathbf{S}_t^{-1}\mathbf{B}_t^\top\Big\|_{\mathrm{F}}^2 \leq \eta^2\|\nabla\ell(\mathbf{W}_t)\|_2^2\Big\|\frac{1}{L\eta}\mathbf{I}_n - \mathbf{B}_t\mathbf{S}_t^{-1}\mathbf{B}_t^\top\Big\|_{\mathrm{F}}^2.
\tag{17}
$$

Again using the definitions of $\Delta\mathbf{A}_t$ and $\Delta\mathbf{B}_t$, the third term in (15) satisfies

$$
\begin{aligned}
\langle\mathbf{A}_t\mathbf{S}_t\Delta\mathbf{B}_t^\top, \Delta\mathbf{A}_t\mathbf{S}_t^{-1}\mathbf{B}_t^\top\rangle_{\mathrm{F}} &= \eta^2\langle\mathbf{A}_t\mathbf{S}_t\mathbf{A}_t^\top\nabla\ell(\mathbf{W}_t), \nabla\ell(\mathbf{W}_t)\mathbf{B}_t\mathbf{S}_t^{-1}\mathbf{B}_t^\top\rangle_{\mathrm{F}} \\
&\overset{(a)}{\leq} \eta^2\|\mathbf{A}_t\mathbf{S}_t\mathbf{A}_t^\top\nabla\ell(\mathbf{W}_t)\|_{\mathrm{F}}\|\nabla\ell(\mathbf{W}_t)\mathbf{B}_t\mathbf{S}_t^{-1}\mathbf{B}_t^\top\|_{\mathrm{F}} \\
&\leq \eta^2\|\nabla\ell(\mathbf{W}_t)\|_2^2\|\mathbf{A}_t\mathbf{S}_t\mathbf{A}_t^\top\|_{\mathrm{F}}\|\mathbf{B}_t\mathbf{S}_t^{-1}\mathbf{B}_t^\top\|_{\mathrm{F}}
\end{aligned}
\tag{18}
$$

where $(a)$ follows from Cauchy-Schwarz inequality.

Regarding the last non-constant term in (15), it holds

$$
-\langle\mathbf{A}_t\mathbf{S}_t\Delta\mathbf{B}_t^\top + \Delta\mathbf{A}_t\mathbf{S}_t^{-1}\mathbf{B}_t^\top, \Delta\mathbf{A}_t\Delta\mathbf{B}_t^\top\rangle_{\mathrm{F}}
$$

$$= \eta^3 \langle \mathbf{A}_t \mathbf{S}_t \mathbf{A}_t^\top \nabla \ell(\mathbf{W}_t) + \nabla \ell(\mathbf{W}_t) \mathbf{B}_t \mathbf{S}_t^{-1} \mathbf{B}_t^\top, \nabla \ell(\mathbf{W}_t) \mathbf{B}_t \mathbf{A}_t^\top \nabla \ell(\mathbf{W}_t) \rangle_{\mathrm{F}}$$

$$\leq \eta^3 \left( \left\| \mathbf{A}_t \mathbf{S}_t \mathbf{A}_t^\top \nabla \ell(\mathbf{W}_t) \right\|_{\mathrm{F}} + \left\| \nabla \ell(\mathbf{W}_t) \mathbf{B}_t \mathbf{S}_t^{-1} \mathbf{B}_t^\top \right\|_{\mathrm{F}} \right) \left\| \nabla \ell(\mathbf{W}_t) \mathbf{B}_t \mathbf{A}_t^\top \nabla \ell(\mathbf{W}_t) \right\|_{\mathrm{F}}$$

$$\leq \eta^3 \| \nabla \ell(\mathbf{W}_t) \|_2^3 \left( \left\| \mathbf{A}_t \mathbf{S}_t \mathbf{A}_t^\top \right\|_{\mathrm{F}} + \left\| \mathbf{B}_t \mathbf{S}_t^{-1} \mathbf{B}_t^\top \right\|_{\mathrm{F}} \right) \| \mathbf{B}_t \mathbf{A}_t^\top \|_{\mathrm{F}} = \mathcal{O}(\eta^3). \quad (19)$$

When performing fine-tuning from a pre-trained weight, both $\eta$ and $\| \nabla \ell(\mathbf{W}_t) \|_2$ are observed to be tiny[1]. As a result, (19) is dominated by (18), and is neglectable in practice; see also [60, 67].

Plugging (16)-(19) into (15) yields

$$\ell(\mathbf{W}_t + \Delta \tilde{\mathbf{W}}_t(\mathbf{S}_t)) \leq \frac{L\eta^2}{2} \| \nabla \ell(\mathbf{W}_t) \|_2^2 \Big[ \Big\| \frac{1}{L\eta} \mathbf{I}_m - \mathbf{A}_t \mathbf{S}_t \mathbf{A}_t^\top \Big\|_{\mathrm{F}}^2 + \Big\| \frac{1}{L\eta} \mathbf{I}_n - \mathbf{B}_t \mathbf{S}_t^{-1} \mathbf{B}_t^\top \Big\|_{\mathrm{F}}^2 +$$

$$2 \| \mathbf{A}_t \mathbf{S}_t \mathbf{A}_t^\top \|_{\mathrm{F}} \| \mathbf{B}_t \mathbf{S}_t^{-1} \mathbf{B}_t^\top \|_{\mathrm{F}} \Big] + \mathcal{O}(L\eta^3) + \text{Const.}$$

$$\stackrel{(a)}{=} \frac{L\eta^2}{2} \| \nabla \ell(\mathbf{W}_t) \|_2^2 \Big[ \| \mathbf{A}_t \mathbf{S}_t \mathbf{A}_t^\top \|_{\mathrm{F}}^2 - \frac{2}{L\eta} \langle \mathbf{I}_m, \mathbf{A}_t \mathbf{S}_t \mathbf{A}_t^\top \rangle_{\mathrm{F}} + \| \mathbf{B}_t \mathbf{S}_t^{-1} \mathbf{B}_t^\top \|_{\mathrm{F}}^2 -$$

$$\frac{2}{L\eta} \langle \mathbf{I}_n, \mathbf{B}_t \mathbf{S}_t^{-1} \mathbf{B}_t^\top \rangle_{\mathrm{F}} + 2 \| \mathbf{A}_t \mathbf{S}_t \mathbf{A}_t^\top \|_{\mathrm{F}} \| \mathbf{B}_t \mathbf{S}_t^{-1} \mathbf{B}_t^\top \|_{\mathrm{F}} \Big] + \mathcal{O}(L\eta^3) + \text{Const.}$$

$$\stackrel{(b)}{=} \frac{L\eta^2}{2} \| \nabla \ell(\mathbf{W}_t) \|_2^2 \Big[ \left( \| \mathbf{A}_t \mathbf{S}_t \mathbf{A}_t^\top \|_{\mathrm{F}} + \| \mathbf{B}_t \mathbf{S}_t^{-1} \mathbf{B}_t^\top \|_{\mathrm{F}} \right)^2 - \frac{2}{L\eta} \times$$

$$\left( \| \mathbf{A}_t \mathbf{S}_t^{\frac{1}{2}} \|_{\mathrm{F}}^2 + \| \mathbf{B}_t \mathbf{S}_t^{-\frac{1}{2}} \|_{\mathrm{F}}^2 \right) \Big] + \mathcal{O}(L\eta^3) + \text{Const.}$$

$$\stackrel{(c)}{\leq} \frac{L\eta^2}{2} \| \nabla \ell(\mathbf{W}_t) \|_2^2 \Big[ \left( \| \mathbf{A}_t \mathbf{S}_t^{\frac{1}{2}} \|_{\mathrm{F}}^2 + \| \mathbf{B}_t \mathbf{S}_t^{-\frac{1}{2}} \|_{\mathrm{F}}^2 \right)^2 - \frac{2}{L\eta} \times$$

$$\left( \| \mathbf{A}_t \mathbf{S}_t^{\frac{1}{2}} \|_{\mathrm{F}}^2 + \| \mathbf{B}_t \mathbf{S}_t^{-\frac{1}{2}} \|_{\mathrm{F}}^2 \right) \Big] + \mathcal{O}(L\eta^3) + \text{Const.}$$

$$\stackrel{(d)}{=} \frac{L\eta^2}{2} \| \nabla \ell(\mathbf{W}_t) \|_2^2 \Big( \| \mathbf{A}_t \mathbf{S}_t^{\frac{1}{2}} \|_{\mathrm{F}}^2 + \| \mathbf{B}_t \mathbf{S}_t^{-\frac{1}{2}} \|_{\mathrm{F}}^2 - \frac{1}{L\eta} \Big)^2 + \mathcal{O}(L\eta^3) + \text{Const.}$$

where $(a)$ expands the squared Frobenius norm; $(b)$ follows from $\langle \mathbf{I}_m, \mathbf{A}_t \mathbf{S}_t \mathbf{A}_t^\top \rangle_{\mathrm{F}} = \mathrm{tr}(\mathbf{A}_t \mathbf{S}_t \mathbf{A}_t^\top) = \| \mathbf{A}_t \mathbf{S}_t^{1/2} \|_{\mathrm{F}}^2$; $(c)$ is because $\mathbf{A}_t \mathbf{S}_t \mathbf{A}_t^\top$ is SPD, thus $\| \mathbf{A}_t \mathbf{S}_t \mathbf{A}_t^\top \|_{\mathrm{F}} = \mathrm{tr}^{1/2}(\mathbf{A}_t \mathbf{S}_t \mathbf{A}_t^\top \mathbf{A}_t \mathbf{S}_t \mathbf{A}_t^\top) = [\sum_i \lambda_i(\mathbf{A}_t \mathbf{S}_t \mathbf{A}_t^\top \mathbf{A}_t \mathbf{S}_t \mathbf{A}_t^\top)]^{1/2} = [\sum_i \lambda_i^2(\mathbf{A}_t \mathbf{S}_t \mathbf{A}_t^\top)]^{1/2} \leq \sum_i \lambda_i(\mathbf{A}_t \mathbf{S}_t \mathbf{A}_t^\top) = \mathrm{tr}(\mathbf{A}_t \mathbf{S}_t \mathbf{A}_t^\top) = \| \mathbf{A}_t \mathbf{S}_t^{1/2} \|_{\mathrm{F}}^2$; and $(d)$ utilizes completing the square. $\qquad \square$

### B.3 Proof of Theorem 3

*Proof.* For SPD matrix $\mathbf{S}_t$, the eigendecomposition gives $\mathbf{S}_t = \mathbf{Q}_t^S \mathrm{diag}(\boldsymbol{\lambda}_t^S) \mathbf{Q}_t^{S\top}$, where $\mathbf{Q}_t^S \in \mathrm{O}(r)$ and $\boldsymbol{\lambda}_t^S$ is element-wise positive. Thus, the objective (8) can be reformulated as

$$\min_{\substack{\mathbf{Q}_t^S \in \mathrm{O}(r) \\ \boldsymbol{\lambda}_t^S \succeq 0}} \left( \| \mathbf{A}_t \mathbf{Q}_t^S \mathrm{diag}^{\frac{1}{2}}(\boldsymbol{\lambda}_t^S) \mathbf{Q}_t^{S\top} \|_{\mathrm{F}}^2 + \| \mathbf{B}_t \mathbf{Q}_t^S \mathrm{diag}^{-\frac{1}{2}}(\boldsymbol{\lambda}_t^S) \mathbf{Q}_t^{S\top} \|_{\mathrm{F}}^2 - \frac{1}{L\eta} \right)^2$$

$$= \min_{\substack{\mathbf{Q}_t^S \in \mathrm{O}(r) \\ \boldsymbol{\lambda}_t^S \succeq 0}} \left( \| \mathbf{A}_t \mathbf{Q}_t^S \mathrm{diag}^{\frac{1}{2}}(\boldsymbol{\lambda}_t^S) \|_{\mathrm{F}}^2 + \| \mathbf{B}_t \mathbf{Q}_t^S \mathrm{diag}^{-\frac{1}{2}}(\boldsymbol{\lambda}_t^S) \|_{\mathrm{F}}^2 - \frac{1}{L\eta} \right)^2$$

$$= \min_{\substack{\mathbf{Q}_t^S \in \mathrm{O}(r) \\ \boldsymbol{\lambda}_t^S \succeq 0}} \left( \sum_{i=1}^r [\boldsymbol{\lambda}_t^S]_i \| \mathbf{A}_t [\mathbf{Q}_t^S]_i \|_2^2 + \sum_{i=1}^r [\boldsymbol{\lambda}_t^S]_i^{-1} \| \mathbf{B}_t [\mathbf{Q}_t^S]_i \|_2^2 - \frac{1}{L\eta} \right)^2 := f(\mathbf{Q}_t^S, \boldsymbol{\lambda}_t^S). \quad (20)$$

Since $f(\mathbf{Q}_t^S, \boldsymbol{\lambda}_t^S)$ is continuous w.r.t. both $\mathbf{Q}_t^S \in \mathrm{O}(r)$ and $\boldsymbol{\lambda}_t^S \succeq 0$, its infimum can be determined by checking the limit of $f$ as it approaches the boundary of the open set $\mathbb{S}_{++}^r$, and analyzing stationary points in the interior [47].

Notice that $\mathrm{O}(r)$ is closed [47], while $(0, +\infty)^r$ is open. It follows from (20) that, for any given $\mathbf{Q}_t^S \in \mathrm{O}(r)$, if some $[\boldsymbol{\lambda}_t^S]_i$ approaches 0 or $+\infty$, the objective $f(\mathbf{Q}_t^S, \boldsymbol{\lambda}_t^S)$ goes to $+\infty$. As a consequence, the minimum must be attained inside the interior of $\mathbb{S}_{++}^r$.

---

[1]Typically, $\eta = \mathcal{O}(10^{-4})$, and $\| \nabla \ell(\mathbf{W}_t) \|_2 \leq \| \nabla \ell(\mathbf{W}_t) \|_{\mathrm{F}} = \mathcal{O}(10^{-1})$ per matrix.

Next, the stationary points are investigated under the following two cases.

**Case 1:** $\eta \geq 1/(\tilde{C}_t L)$ or $\eta < 0$.

Defining $g(\mathbf{S}_t) := \|\mathbf{A}_t \mathbf{S}_t^{\frac{1}{2}}\|_{\mathrm{F}}^2 + \|\mathbf{B}_t \mathbf{S}_t^{-\frac{1}{2}}\|_{\mathrm{F}}^2$, it will be shown that $\min_{\mathbf{S}_t \in \mathbb{S}_{++}^r} g(\mathbf{S}_t) = \tilde{C}_t$ and the corresponding minimizer is uniquely $\tilde{\mathbf{S}}_t$.

From the stationary point condition we have

$$
\begin{aligned}
\nabla g(\mathbf{S}_t) &= \mathbf{A}_t^\top \mathbf{A}_t - \mathbf{S}_t^{-1} \mathbf{B}_t^\top \mathbf{B}_t \mathbf{S}_t^{-1} = \mathbf{0} \\
\Rightarrow\ & \mathbf{S}_t \mathbf{A}_t^\top \mathbf{A}_t \mathbf{S}_t = \mathbf{B}_t^\top \mathbf{B}_t \\
\Rightarrow\ & (\mathbf{A}_t^\top \mathbf{A}_t)^{\frac{1}{2}} \mathbf{S}_t (\mathbf{A}_t^\top \mathbf{A}_t)^{\frac{1}{2}} (\mathbf{A}_t^\top \mathbf{A}_t)^{\frac{1}{2}} \mathbf{S}_t (\mathbf{A}_t^\top \mathbf{A}_t)^{\frac{1}{2}} = (\mathbf{A}_t^\top \mathbf{A}_t)^{\frac{1}{2}} \mathbf{B}_t^\top \mathbf{B}_t^\top (\mathbf{A}_t^\top \mathbf{A}_t)^{\frac{1}{2}} \\
\Rightarrow\ & \big[ (\mathbf{A}_t^\top \mathbf{A}_t)^{\frac{1}{2}} \mathbf{S}_t (\mathbf{A}_t^\top \mathbf{A}_t)^{\frac{1}{2}} \big]^2 = (\mathbf{A}_t^\top \mathbf{A}_t)^{\frac{1}{2}} \mathbf{B}_t^\top \mathbf{B}_t^\top (\mathbf{A}_t^\top \mathbf{A}_t)^{\frac{1}{2}}.
\end{aligned}
\tag{21}
$$

Since $(\mathbf{A}_t^\top \mathbf{A}_t)^{\frac{1}{2}} \mathbf{S}_t (\mathbf{A}_t^\top \mathbf{A}_t)^{\frac{1}{2}}$ is also SPD when $\mathbf{S}_t \in \mathbb{S}_{++}^r$ and $\mathrm{rank}(\mathbf{A}_t) = r$, its solution is uniquely given by the positive square root

$$
(\mathbf{A}_t^\top \mathbf{A}_t)^{\frac{1}{2}} \mathbf{S}_t (\mathbf{A}_t^\top \mathbf{A}_t)^{\frac{1}{2}} = \big[ (\mathbf{A}_t^\top \mathbf{A}_t)^{\frac{1}{2}} \mathbf{B}_t^\top \mathbf{B}_t^\top (\mathbf{A}_t^\top \mathbf{A}_t)^{\frac{1}{2}} \big]^{\frac{1}{2}}.
$$

Left- and right-multiplying $(\mathbf{A}_t^\top \mathbf{A}_t)^{-1/2}$ results in

$$
\mathbf{S}_t = (\mathbf{A}_t^\top \mathbf{A}_t)^{-\frac{1}{2}} \big[ (\mathbf{A}_t^\top \mathbf{A}_t)^{\frac{1}{2}} \mathbf{B}_t^\top \mathbf{B}_t (\mathbf{A}_t^\top \mathbf{A}_t)^{\frac{1}{2}} \big]^{\frac{1}{2}} (\mathbf{A}_t^\top \mathbf{A}_t)^{-\frac{1}{2}} = \tilde{\mathbf{S}}_t.
$$

Given that $g(\mathbf{S}_t)$ approaches $+\infty$ on the boundary of $\mathbb{S}_{++}^r$ and has only one stationary point $\tilde{\mathbf{S}}_t$, its minimum is reached uniquely at $\tilde{\mathbf{S}}_t$, i.e.,

$$
\begin{aligned}
\min_{\mathbf{S}_t \in \mathbb{S}_{++}^r} g(\mathbf{S}_t) &= \|\mathbf{A}_t \tilde{\mathbf{S}}_t^{\frac{1}{2}}\|_{\mathrm{F}}^2 + \|\mathbf{B}_t \tilde{\mathbf{S}}_t^{-\frac{1}{2}}\|_{\mathrm{F}}^2 = \mathrm{tr}(\mathbf{A}_t \tilde{\mathbf{S}}_t \mathbf{A}_t^\top) + \mathrm{tr}(\mathbf{B}_t \tilde{\mathbf{S}}_t^{-1} \mathbf{B}_t^\top) \\
&\overset{(a)}{=} \mathrm{tr}(\mathbf{A}_t^\top \mathbf{A}_t \tilde{\mathbf{S}}_t) + \mathrm{tr}(\tilde{\mathbf{S}}_t^{-1} \mathbf{B}_t^\top \mathbf{B}_t) \overset{(b)}{=} 2\,\mathrm{tr}(\mathbf{A}_t^\top \mathbf{A}_t \tilde{\mathbf{S}}_t) \\
&= 2\|\mathbf{A}_t \tilde{\mathbf{S}}_t^{\frac{1}{2}}\|_{\mathrm{F}}^2 = 2\|\mathbf{B}_t \tilde{\mathbf{S}}_t^{-\frac{1}{2}}\|_{\mathrm{F}}^2
\end{aligned}
\tag{22}
$$

where $(a)$ relies on the cyclic property of trace, and $(b)$ utilizes (21).

Next, we prove the minimum value in (22) can be equivalently expressed as $\|\mathbf{A}_t \mathbf{B}_t^\top\|_*$.

Let $\mathbf{A}_t = \mathbf{U}_t^A \mathbf{\Sigma}_t^A \mathbf{V}_t^{A\top}$ be the economy-sized SVD. By the definition of $\tilde{\mathbf{S}}_t$, it follows that

$$
\begin{aligned}
\|\mathbf{A}_t \tilde{\mathbf{S}}_t^{\frac{1}{2}}\|_{\mathrm{F}}^2 &= \mathrm{tr}\big( \mathbf{A}_t (\mathbf{A}_t^\top \mathbf{A}_t)^{-\frac{1}{2}} \big[ (\mathbf{A}_t^\top \mathbf{A}_t)^{\frac{1}{2}} \mathbf{B}_t^\top \mathbf{B}_t (\mathbf{A}_t^\top \mathbf{A}_t)^{\frac{1}{2}} \big]^{\frac{1}{2}} (\mathbf{A}_t^\top \mathbf{A}_t)^{-\frac{1}{2}} \mathbf{A}_t^\top \big) \\
&\overset{(a)}{=} \mathrm{tr}\big( \big[ (\mathbf{A}_t^\top \mathbf{A}_t)^{\frac{1}{2}} \mathbf{B}_t^\top \mathbf{B}_t (\mathbf{A}_t^\top \mathbf{A}_t)^{\frac{1}{2}} \big]^{\frac{1}{2}} \big) = \sum_{i=1}^r \lambda_i \big( \big[ (\mathbf{A}_t^\top \mathbf{A}_t)^{\frac{1}{2}} \mathbf{B}_t^\top \mathbf{B}_t (\mathbf{A}_t^\top \mathbf{A}_t)^{\frac{1}{2}} \big]^{\frac{1}{2}} \big) \\
&= \sum_{i=1}^r \lambda_i^{\frac{1}{2}} \big( (\mathbf{A}_t^\top \mathbf{A}_t)^{\frac{1}{2}} \mathbf{B}_t^\top \mathbf{B}_t (\mathbf{A}_t^\top \mathbf{A}_t)^{\frac{1}{2}} \big) = \sum_{i=1}^r \sigma_i \big( (\mathbf{A}_t^\top \mathbf{A}_t)^{\frac{1}{2}} \mathbf{B}_t^\top \big) \\
&= \sum_{i=1}^r \sigma_i \big( \mathbf{V}_t^A \mathbf{\Sigma}_t^A \mathbf{V}_t^{A\top} \mathbf{B}_t^\top \big) = \sum_{i=1}^r \sigma_i \big( \mathbf{U}_t^A \mathbf{\Sigma}_t^A \mathbf{V}_t^{A\top} \mathbf{B}_t^\top \big) \\
&= \sum_{i=1}^r \sigma_i \big( \mathbf{A}_t \mathbf{B}_t^\top \big) \overset{(b)}{=} \|\mathbf{A}_t \mathbf{B}_t^\top\|_*
\end{aligned}
$$

where $(a)$ relies on the cyclic property of trace, and $(b)$ is from the definition of nuclear norm.

From the condition $\eta \in (-\infty, 0) \cup (1/(\tilde{C}_t L), +\infty)$, we have $g(\mathbf{S}_t) - 1/(L\eta) \geq 0$, $\forall \mathbf{S}_t \in \mathbb{S}_{++}^r$, thus

$$
\underset{\mathbf{S}_t \in \mathbb{S}_{++}^r}{\arg\min} \left( \|\mathbf{A}_t \mathbf{S}_t^{\frac{1}{2}}\|_{\mathrm{F}}^2 + \|\mathbf{B}_t \mathbf{S}_t^{-\frac{1}{2}}\|_{\mathrm{F}}^2 - \frac{1}{L\eta} \right)^2 = \underset{\mathbf{S}_t \in \mathbb{S}_{++}^r}{\arg\min} \left( g(\mathbf{S}_t) - \frac{1}{L\eta} \right)^2 = \underset{\mathbf{S}_t \in \mathbb{S}_{++}^r}{\arg\min}\, g(\mathbf{S}_t) - \frac{1}{L\eta} = \tilde{\mathbf{S}}_t.
$$

**Case 2:** $0 < \eta < 1/(\tilde{C}_t L)$.

For this case, it holds $\min_{\mathbf{S}_t \in \mathbb{S}_{++}^r} g(\mathbf{S}_t) - 1/(L\eta) < 0$, hence the minimum of (8) is reached when $g(\mathbf{S}_t) = 1/(L\eta)$. In particular, for any $\mathbf{S}_t$ satisfying $g(\mathbf{S}_t) < 1/(L\eta)$, it is possible to find a scaling factor $\gamma_t > 0$ by solving a quadratic equation such that $g(\gamma_t \mathbf{S}_t) = 1/(L\eta)$.

As an example, we consider scaling $\tilde{\mathbf{S}}_t$ to attain $g(\gamma_t \tilde{\mathbf{S}}_t) = 1/(L\eta)$. It is worth emphasizing that, for any choice $\eta \in (0, 1/(\tilde{C}_t L))$, this solution is universally valid because $g(\tilde{\mathbf{S}}_t) < 1/(L\eta)$ always hold in this case.

By the definition of $g(\mathbf{S}_t)$, the sought $g(\gamma_t \tilde{\mathbf{S}}_t) = 1/(L\eta)$ is equivalent to the quadratic equation

$$\frac{1}{L\eta} = \gamma_t \|\mathbf{A}_t \tilde{\mathbf{S}}_t^{\frac{1}{2}}\|_{\mathrm{F}}^2 + \gamma_t^{-1} \|\mathbf{B}_t \mathbf{S}_t^{-\frac{1}{2}}\|_{\mathrm{F}}^2, \ \gamma_t > 0.$$

Solving this equation gives

$$\gamma_t = \frac{\frac{1}{L\eta} \pm \sqrt{\frac{1}{L^2\eta^2} - 4\|\mathbf{A}_t \tilde{\mathbf{S}}_t^{\frac{1}{2}}\|_{\mathrm{F}}^2 \|\mathbf{B}_t \tilde{\mathbf{S}}_t^{-\frac{1}{2}}\|_{\mathrm{F}}^2}}{2\|\mathbf{A}_t \tilde{\mathbf{S}}_t^{\frac{1}{2}}\|_{\mathrm{F}}^2} = \frac{1}{\tilde{C}_t L\eta} \pm \sqrt{\frac{1}{\tilde{C}_t^2 L^2\eta^2} - 1}$$

which concludes the proof. $\qquad \square$

### B.4 Proof of Theorem 5

*Proof.* Plugging $\mathbf{S}_t = s_t \mathbf{I}_r$, $s_t \in \mathbb{R}_{++}$ into (8) incurs alternative objective

$$\min_{s_t \in \mathbb{R}_{++}} h(s_t) := \left( \|\mathbf{A}_t\|_{\mathrm{F}}^2 s_t + \|\mathbf{B}_t\|_{\mathrm{F}}^2 s_t^{-1} - \frac{1}{L\eta} \right)^2. \tag{23}$$

To solve this quartic optimization problem, we consider the following two cases.

**Case 1:** $\eta \geq 1/(2\|\mathbf{A}_t\|_{\mathrm{F}}\|\mathbf{B}_t\|_{\mathrm{F}} L)$ or $\eta < 0$.

In this case we have

$$\|\mathbf{A}_t\|_{\mathrm{F}}^2 s_t + \|\mathbf{B}_t\|_{\mathrm{F}}^2 s_t^{-1} \geq 2\|\mathbf{A}_t\|_{\mathrm{F}}\|\mathbf{B}_t\|_{\mathrm{F}} \geq \frac{1}{L\eta}.$$

Since $(\cdot)^2$ monotonically increases on $\mathbb{R}_+$, the unique global optimum of (23) is

$$s_t^* = \arg\min_{s_t \in \mathbb{R}_{++}} \|\mathbf{A}_t\|_{\mathrm{F}}^2 s_t + \|\mathbf{B}_t\|_{\mathrm{F}}^2 s_t^{-1} - \frac{1}{L\eta} = \frac{\|\mathbf{B}_t\|_{\mathrm{F}}}{\|\mathbf{A}_t\|_{\mathrm{F}}}.$$

**Case 2:** $0 < \eta < 1/(2\|\mathbf{A}_t\|_{\mathrm{F}}\|\mathbf{B}_t\|_{\mathrm{F}} L)$.

The proof for this case relies on Descartes' rule of signs; cf. Theorem 8.

The gradients of the objective function $h$ is given by

$$h'(s_t) = 2\|\mathbf{A}_t\|_{\mathrm{F}}^4 s_t - 2\|\mathbf{B}_t\|_{\mathrm{F}}^4 s_t^{-3} - \frac{2}{L\eta}\|\mathbf{A}_t\|_{\mathrm{F}}^2 + \frac{2}{L\eta}\|\mathbf{B}_t\|_{\mathrm{F}}^2 s_t^{-2}$$

$$= 2s_t^{-3}\left( \|\mathbf{A}_t\|_{\mathrm{F}}^4 s_t^4 - \frac{1}{L\eta}\|\mathbf{A}_t\|_{\mathrm{F}}^2 s_t^3 + \frac{2}{L\eta}\|\mathbf{B}_t\|_{\mathrm{F}}^2 s_t - 2\|\mathbf{B}_t\|_{\mathrm{F}}^4 \right) \tag{24}$$

where the quartic polynomial in the parenthesis has coefficients with signs $(+, -, +, -)$. Using Theorem 8, the gradient (24) has 3 or 1 positive roots. That says, (23) has either 3 or 1 stationary point(s).

Notice that the objective (23) must be non-negative, and its lower bound of 0 can be reached if and only if $\|\mathbf{A}_t\|_{\mathrm{F}}^2 s_t + \|\mathbf{B}_t\|_{\mathrm{F}}^2 s_t^{-1} - 1/(L\eta) = 0$. Solving this quadratic equation over $s_t \in \mathbb{R}_{++}$ leads to two global minimum

$$s_t^* = \frac{\frac{1}{L\eta} \pm \sqrt{\frac{1}{L^2\eta^2} - 4\|\mathbf{A}_t\|_{\mathrm{F}}^2 \|\mathbf{B}_t\|_{\mathrm{F}}^2}}{2\|\mathbf{A}_t\|_{\mathrm{F}}^2}. \tag{25}$$

Hence, (23) must have 3 stationary points. As $\min_{s_t > 0} \|\mathbf{A}_t\|_{\mathrm{F}}^2 s_t + \|\mathbf{B}_t\|_{\mathrm{F}}^2 s_t^{-1} = 2\|\mathbf{A}_t\|_{\mathrm{F}}\|\mathbf{B}_t\|_{\mathrm{F}} < 1/(L\eta)$, the remaining stationary point is a local maximum at $s_t = \|\mathbf{B}_t\|_{\mathrm{F}}/\|\mathbf{A}_t\|_{\mathrm{F}}$.

Lastly, the objective (23) is a continuous function of $s_t \in (0, +\infty)$, and tends to $+\infty$ when $s_t \to 0$ and $s_t \to +\infty$. We conclude it only has two global minimum given by (25). $\qquad \square$

## B.5 Proof of Theorem 4

*Proof.* By the definition (5) of $\tilde{\mathbf{A}}_t$ and $\tilde{\mathbf{B}}_t$, it holds

$$\langle\nabla_{\tilde{\mathbf{A}}_t}\ell(\tilde{\mathbf{W}}_t),\Delta\tilde{\mathbf{A}}_t\rangle_{\mathrm{F}} = \langle\nabla\ell(\mathbf{W}_t)\tilde{\mathbf{B}}_t,-\eta\nabla\ell(\mathbf{W}_t)\tilde{\mathbf{B}}_t\rangle_{\mathrm{F}} = \eta\langle\nabla\ell(\mathbf{W}_t)\tilde{\mathbf{B}}_t,\nabla\ell(\mathbf{W}_t)\tilde{\mathbf{B}}_t\rangle_{\mathrm{F}}$$
$$= -\eta\big\|\nabla\ell(\mathbf{W}_t)\mathbf{B}_t\mathbf{S}_t^{-\frac{1}{2}}\big\|_{\mathrm{F}}^2 \leq 0$$

Likewise, we have

$$\langle\nabla_{\mathbf{B}_t}\ell(\mathbf{W}_t),\Delta\tilde{\mathbf{B}}_t\rangle_{\mathrm{F}} = -\eta\big\|\nabla\ell(\mathbf{W}_t)^\top\mathbf{A}_t\mathbf{S}_t^{\frac{1}{2}}\big\|_{\mathrm{F}}^2 \leq 0.$$

As a consequence,

$$\langle\nabla_{\tilde{\mathbf{A}}_t}\ell(\tilde{\mathbf{W}}_t),\Delta\tilde{\mathbf{A}}_t\rangle_{\mathrm{F}} + \langle\nabla_{\tilde{\mathbf{B}}_t}\ell(\tilde{\mathbf{W}}_t),\Delta\tilde{\mathbf{B}}_t\rangle_{\mathrm{F}} = -\eta\big(\|\nabla\ell(\mathbf{W}_t)\mathbf{B}_t\mathbf{S}_t^{-\frac{1}{2}}\|_{\mathrm{F}}^2 + \|\nabla\ell(\mathbf{W}_t)^\top\mathbf{A}_t\mathbf{S}_t^{\frac{1}{2}}\|_{\mathrm{F}}^2\big)$$
$$\geq \|\nabla\ell(\mathbf{W}_t)\|_2^2\big(\|\mathbf{A}_t\mathbf{S}_t^{\frac{1}{2}}\|_{\mathrm{F}}^2 + \|\mathbf{B}_t\mathbf{S}_t^{-\frac{1}{2}}\|_{\mathrm{F}}^2\big).$$

It follows from (22) that $\mathbf{S}_t = \tilde{\mathbf{S}}_t$ is the unique global minimum of this lower bound. $\qquad\square$

## B.6 Proof of Theorem 6

*Proof.* As $\tilde{\mathbf{A}}_t'\tilde{\mathbf{B}}_t'^\top = \mathbf{A}_t'\mathbf{B}_t'^\top = \mathbf{A}_t\mathbf{B}_t^\top = \tilde{\mathbf{A}}_t\tilde{\mathbf{B}}_t^\top$, Lemma 1 suggests there exists $\mathbf{Q}_t \in \mathrm{GL}(r)$ such that $\tilde{\mathbf{A}}_t' = \tilde{\mathbf{A}}_t\mathbf{Q}_t$ and $\tilde{\mathbf{B}}_t' = \tilde{\mathbf{B}}_t\mathbf{Q}_t^{-\top}$. Further, to prove Theorem 6, it is sufficient to prove that $\mathbf{Q}_t \in \mathrm{O}(r)$. Since $\tilde{\mathbf{A}}_t$ has full rank, the condition $\mathbf{Q}_t \in \mathrm{O}(r)$ is equivalent to $\tilde{\mathbf{A}}_t'\tilde{\mathbf{A}}_t'^\top = \tilde{\mathbf{A}}_t\tilde{\mathbf{A}}_t^\top$.

By Lemma 7, this can be simplify as

$$\tilde{\mathbf{A}}_t'\tilde{\mathbf{A}}_t'^\top = \mathbf{A}_t'\mathbf{S}_t'\mathbf{A}_t'^\top = \mathbf{A}_t'(\mathbf{A}_t'^\top\mathbf{A}_t')^{-1}(\mathbf{A}_t'^\top\mathbf{A}_t'\mathbf{B}_t'^\top\mathbf{B}_t')^{\frac{1}{2}}\mathbf{A}_t'^\top$$

Again using Lemma 1 and that $\mathbf{A}_t'\mathbf{B}_t'^\top = \mathbf{A}_t\mathbf{B}_t^\top$, there exists $\mathbf{P}_t \in \mathrm{GL}(r)$ such that $\mathbf{A}_t' = \mathbf{A}_t\mathbf{P}_t$ and $\mathbf{B}_t' = \mathbf{B}_t\mathbf{P}_t^{-\top}$, which yields

$$\tilde{\mathbf{A}}_t'\tilde{\mathbf{A}}_t'^\top = \mathbf{A}_t\mathbf{P}_t(\mathbf{P}_t^\top\mathbf{A}_t^\top\mathbf{A}_t\mathbf{P}_t)^{-1}(\mathbf{P}_t^\top\mathbf{A}_t^\top\mathbf{A}_t\mathbf{P}_t\mathbf{P}_t^{-1}\mathbf{B}_t^\top\mathbf{B}_t\mathbf{P}_t^{-\top})^{\frac{1}{2}}\mathbf{P}_t^\top\mathbf{A}_t^\top$$
$$= \mathbf{A}_t(\mathbf{A}_t^\top\mathbf{A}_t)^{-1}\mathbf{P}_t^{-\top}(\mathbf{P}_t^\top\mathbf{A}_t^\top\mathbf{A}_t\mathbf{B}_t^\top\mathbf{B}_t\mathbf{P}_t^{-\top})^{\frac{1}{2}}\mathbf{P}_t^\top\mathbf{A}_t^\top$$

Notice that

$$\big[\mathbf{P}_t^{-\top}(\mathbf{P}_t^\top\mathbf{A}_t^\top\mathbf{A}_t\mathbf{B}_t^\top\mathbf{B}_t\mathbf{P}_t^{-\top})^{\frac{1}{2}}\mathbf{P}_t^\top\big]^2 = \mathbf{P}_t^{-\top}(\mathbf{P}_t^\top\mathbf{A}_t^\top\mathbf{A}_t\mathbf{B}_t^\top\mathbf{B}_t\mathbf{P}_t^{-\top})\mathbf{P}_t^\top = \mathbf{A}_t^\top\mathbf{A}_t\mathbf{B}_t^\top\mathbf{B}_t$$
$$\implies \mathbf{P}_t^{-\top}(\mathbf{P}_t^\top\mathbf{A}_t^\top\mathbf{A}_t\mathbf{B}_t^\top\mathbf{B}_t\mathbf{P}_t^{-\top})^{\frac{1}{2}}\mathbf{P}_t^\top = (\mathbf{A}_t^\top\mathbf{A}_t\mathbf{B}_t^\top\mathbf{B}_t)^{\frac{1}{2}}.$$

It then follows

$$\tilde{\mathbf{A}}_t'\tilde{\mathbf{A}}_t'^\top = \mathbf{A}_t(\mathbf{A}_t^\top\mathbf{A}_t)^{-1}(\mathbf{A}_t^\top\mathbf{A}_t\mathbf{B}_t^\top\mathbf{B}_t)^{\frac{1}{2}}\mathbf{A}_t^\top \stackrel{(a)}{=} \mathbf{A}_t\tilde{\mathbf{S}}_t\mathbf{A}_t^\top = \tilde{\mathbf{A}}_t\tilde{\mathbf{A}}_t^\top$$

where $(a)$ applies (26) with $\mathbf{X} = \mathbf{A}_t^\top\mathbf{A}_t$ and $\mathbf{Y} = \mathbf{B}_t^\top\mathbf{B}_t$.

The proof is thereby completed. $\qquad\square$

## B.7 Proof for Riemannian metric and gradient

We first show that the Riemannian metric (12) leads to update 11. The subscript $t$ is dropped for simplicity. By definition, the Riemannian gradient $(\mathbf{G}_A,\mathbf{G}_B)$ should satisfy

$$g_{(\mathbf{A},\mathbf{B})}((\mathbf{G}_A,\mathbf{G}_B),(\mathbf{Z}_A,\mathbf{Z}_B)) = \langle\nabla_\mathbf{A}\ell(\mathbf{W}),\mathbf{Z}_A\rangle_{\mathrm{F}} + \langle\nabla_\mathbf{B}\ell(\mathbf{W}),\mathbf{Z}_B\rangle_{\mathrm{F}}, \forall\mathbf{Z}_A \in \mathbb{R}^{m\times r}, \mathbf{Z}_B \in \mathbb{R}^{n\times r}.$$

Using (12), the Riemannian gradient is thus given by

$$\langle\mathbf{G}_A\tilde{\mathbf{S}},\mathbf{Z}_A\rangle_{\mathrm{F}} = \langle\nabla_\mathbf{A}\ell(\mathbf{W}),\mathbf{Z}_A\rangle_{\mathrm{F}}, \forall\mathbf{Z}_A \Rightarrow \mathbf{G}_A = \nabla_\mathbf{A}\ell(\mathbf{W})\tilde{\mathbf{S}}^{-1} = \nabla\ell(\mathbf{W})\mathbf{B}\tilde{\mathbf{S}}^{-1},$$
$$\langle\mathbf{G}_B\tilde{\mathbf{S}}^{-1},\mathbf{Z}_B\rangle_{\mathrm{F}} = \langle\nabla_\mathbf{B}\ell(\mathbf{W}),\mathbf{Z}_B\rangle_{\mathrm{F}}, \forall\mathbf{Z}_B \Rightarrow \mathbf{G}_A = \nabla_\mathbf{B}\ell(\mathbf{W})\tilde{\mathbf{S}} = \nabla\ell(\mathbf{W})^\top\mathbf{A}\tilde{\mathbf{S}}.$$

We next show that our metric (12) guarantees the update of $(\mathbf{A},\mathbf{B})$ is always on the horizontal space, and has no component on the vertical space. First, the vertical space at $(\mathbf{A},\mathbf{B})$ is $T_{(\mathbf{A},\mathbf{B})}^{\mathrm{vert}} = \{(\mathbf{A}\mathbf{X},-\mathbf{B}\mathbf{X}^\top) \mid \mathbf{X} \in \mathbb{R}^{r\times r}\}$, which can be easily verified via

$$(\mathbf{A} + \epsilon\mathbf{A}\mathbf{X})(\mathbf{B} - \epsilon\mathbf{B}\mathbf{X}^\top)^\top = \mathbf{A}\mathbf{B}^\top + \mathcal{O}(\epsilon^2).$$

Thus, for update $(\Delta\mathbf{A}, \Delta\mathbf{B})$ to be purely horizontal, we must have

$$g_{(\mathbf{A},\mathbf{B})}((\mathbf{A}\mathbf{X}, -\mathbf{B}\mathbf{X}^\top), (\Delta\mathbf{A}, \Delta\mathbf{B})) = 0, \ \forall \mathbf{X} \in \mathbb{R}^{r\times r}.$$

From (11), plugging in the update $\Delta\mathbf{A} = -\eta\nabla\ell(\mathbf{W})\mathbf{B}\tilde{\mathbf{S}}^{-1}$ and $\Delta\mathbf{B} = -\eta\nabla\ell(\mathbf{W})^\top\mathbf{B}\tilde{\mathbf{S}}_t$ gives

$$\begin{aligned}
&g_{(\mathbf{A},\mathbf{B})}((\mathbf{A}\mathbf{X}, -\mathbf{B}\mathbf{X}^\top), (\Delta\mathbf{A}, \Delta\mathbf{B})) \\
&= \langle \mathbf{A}\mathbf{X}\tilde{\mathbf{S}}, -\eta\nabla\ell(\mathbf{W})\mathbf{B}\tilde{\mathbf{S}}^{-1}\rangle_\mathrm{F} + \langle -\mathbf{B}\mathbf{X}^\top\tilde{\mathbf{S}}^{-1}, -\eta\nabla\ell(\mathbf{W})^\top\mathbf{A}\tilde{\mathbf{S}}\rangle_\mathrm{F} \\
&= 0
\end{aligned}$$

which completes the proof.

### B.8 Useful facts

**Lemma 7.** *Under Assumption 1, it holds*

$$\tilde{\mathbf{S}}_t = (\mathbf{A}_t^\top\mathbf{A}_t)^{-\frac{1}{2}}\left[(\mathbf{A}_t^\top\mathbf{A}_t)^{\frac{1}{2}}\mathbf{B}_t^\top\mathbf{B}_t(\mathbf{A}_t^\top\mathbf{A}_t)^{\frac{1}{2}}\right]^{\frac{1}{2}}(\mathbf{A}_t^\top\mathbf{A}_t)^{-\frac{1}{2}} = (\mathbf{A}_t^\top\mathbf{A}_t)^{-1}\left[\mathbf{A}_t^\top\mathbf{A}_t\mathbf{B}_t^\top\mathbf{B}_t\right]^{\frac{1}{2}}.$$

*Proof.* It holds for any $\mathbf{X}, \mathbf{Y} \in \mathbb{S}_{++}^r$ that

$$\left[\mathbf{X}^{\frac{1}{2}}(\mathbf{X}^{\frac{1}{2}}\mathbf{Y}\mathbf{X}^{\frac{1}{2}})^{\frac{1}{2}}\mathbf{X}^{-\frac{1}{2}}\right]^2 = \mathbf{X}^{\frac{1}{2}}(\mathbf{X}^{\frac{1}{2}}\mathbf{Y}\mathbf{X}^{\frac{1}{2}})\mathbf{X}^{-\frac{1}{2}} = \mathbf{X}\mathbf{Y}.$$

Taking square root on both sides and left-multiplying $\mathbf{X}^{-1}$ give

$$\mathbf{X}^{-\frac{1}{2}}(\mathbf{X}^{\frac{1}{2}}\mathbf{Y}\mathbf{X}^{\frac{1}{2}})^{\frac{1}{2}}\mathbf{X}^{-\frac{1}{2}} = \mathbf{X}^{-1}(\mathbf{X}\mathbf{Y})^{\frac{1}{2}}. \tag{26}$$

Letting $\mathbf{X} = \mathbf{A}_t^\top\mathbf{A}_t$ and $\mathbf{Y} = \mathbf{B}_t^\top\mathbf{B}_t$ in (26) completes the proof. □

**Theorem 8** (Descartes' rule of signs [11]). *The number of strictly positive roots (counting multiplicity) of polynomial $h$ is equal to the number of sign changes in the coefficients of $h$, minus a nonnegative even number.*

## C  Algorithm pseudocodes

The Appendix provides the step-by-step pseudocodes for RefLoRA and its lightweight version RefLoRA-S. As our algorithms rely on Assumption 1 but LoRA intialize $\mathbf{B}_0 = \mathbf{0}$ by default, one can either utilizes other full-rank initialization schemes [41, 34, 60], or warm up the algorithm for $t_w > 0$ steps until $\mathbf{A}_t$ and $\mathbf{B}_t$ both satisfy full column rank. Without loss of generality, our pseudocodes assume $\mathbf{A}_0$ and $\mathbf{B}_0$ are already full-rank, and define $\mathbf{W}_t := \mathbf{W}^{\mathrm{pt}} + \mathbf{A}_t\mathbf{B}_t^\top == \mathbf{W}^{\mathrm{pt}} + \tilde{\mathbf{A}}_t\tilde{\mathbf{B}}_t^\top$.

For RefLoRA-S, it is worth noting that adaptive optimizers rely on the first and second moments of gradients $\nabla_{\tilde{\mathbf{A}}_t}\ell(\mathbf{W}_t) = \nabla\ell(\mathbf{W}_t)\tilde{\mathbf{B}}_t = \frac{1}{\sqrt{s_t}}\nabla_{\mathbf{A}_t}\ell(\mathbf{W}_t)$ and $\nabla_{\tilde{\mathbf{B}}_t}\ell(\mathbf{W}_t) = \nabla\ell(\mathbf{W}_t)^\top\tilde{\mathbf{A}}_t = \sqrt{s_t}\nabla_{\mathbf{B}_t}\ell(\mathbf{W}_t)$. As a consequence, one can correspondingly scale the gradient moments of $\mathbf{A}_t$ and $\mathbf{B}_t$ to render the gradient moments of $\tilde{\mathbf{A}}_t$ and $\tilde{\mathbf{B}}_t$, thus removing the need for preconditioning.

## D  Experimental setups and hyperparameters

This appendix provides the detailed setups as well as hyperparameters used in our numerical tests.

### D.1  Platforms

All the experiments are conducted on a desktop equipped with an NVIDIA RTX A5000 GPU, and a server with NVIDIA A40 and A100 GPUs. The codes for synthetical tests are written with MATLAB, and codes for LLM-related experiments are in PyTorch. In addition, our implementation of LLM tests are based on [24, 70].

---

**Algorithm 1:** Refactored low-rank adaptation (RefLoRA)

---

**Input:** Loss $\ell$, pre-trained weight $\mathbf{W}^{\mathrm{pt}}$, maximum iterations $T$, and learning rate $\eta$.
**Initialize:** full-rank $\mathbf{A}_0$ and $\mathbf{B}_0$.

1 **for** $t = 0, \ldots, T-1$ **do**
2      Compute $\tilde{\mathbf{S}}_t = (\mathbf{A}_t^\top \mathbf{A}_t)^{-\frac{1}{2}} \left[ (\mathbf{A}_t^\top \mathbf{A}_t)^{\frac{1}{2}} \mathbf{B}_t^\top \mathbf{B}_t (\mathbf{A}_t^\top \mathbf{A}_t)^{\frac{1}{2}} \right]^{\frac{1}{2}} (\mathbf{A}_t^\top \mathbf{A}_t)^{-\frac{1}{2}}$;
3      **if** *adaptive optimizer* **then**
4          Precondition gradients $\mathbf{G}_A = \nabla\ell(\mathbf{W}_t)\mathbf{B}_t\tilde{\mathbf{S}}_t^{-1}$, $\mathbf{G}_B = \nabla\ell(\mathbf{W}_t)^\top \mathbf{A}_t\tilde{\mathbf{S}}_t$;
5          Update $\mathbf{A}_{t+1} = \mathrm{AdaptOpt}(\mathbf{A}_t, \eta, \mathbf{G}_A, t)$, $\mathbf{B}_{t+1} = \mathrm{AdaptOpt}(\mathbf{B}_t, \eta, \mathbf{G}_B, t)$;
6      **else**
7          Refactor $\tilde{\mathbf{A}}_t = \mathbf{A}_t\tilde{\mathbf{S}}^{1/2}$, $\tilde{\mathbf{B}}_t = \mathbf{B}_t\tilde{\mathbf{S}}^{-1/2}$;
8          Update $\mathbf{A}_{t+1} = \tilde{\mathbf{A}}_t - \eta\nabla\ell(\mathbf{W}_t)\tilde{\mathbf{B}}_t$, $\mathbf{B}_{t+1} = \tilde{\mathbf{B}}_t - \eta\nabla\ell(\mathbf{W}_t)^\top \tilde{\mathbf{A}}_t$;
9      **end**
10 **end**
**Output:** $\mathbf{A}_T$ and $\mathbf{B}_T$.

---

---

**Algorithm 2:** Low-rank adaptation with simplified refactoring (RefLoRA-S)

---

**Input:** Loss $\ell$, pre-trained weight $\mathbf{W}^{\mathrm{pt}}$, maximum iterations $T$, and learning rate $\eta$.
**Initialize:** full-rank $\mathbf{A}_0$ and $\mathbf{B}_0$.

1 **for** $t = 0, \ldots, T-1$ **do**
2      Compute $\tilde{s}_t = \|\mathbf{B}_t\|_{\mathrm{F}}/\|\mathbf{A}_t\|_{\mathrm{F}}$;
3      Refactor $\tilde{\mathbf{A}}_t = \sqrt{\tilde{s}}\mathbf{A}_t$, $\tilde{\mathbf{B}}_t = 1/\sqrt{\tilde{s}}\mathbf{B}_t$;
4      **if** *adaptive optimizer* **then**
5          Update $\mathbf{A}_{t+1} = \mathrm{AdaptOpt}(\tilde{\mathbf{A}}_t, \eta, \nabla\ell(\mathbf{W}_t)\tilde{\mathbf{B}}_t, t)$ with first and second moments scaled
         by $1/\sqrt{\tilde{s}}$ and $1/\tilde{s}_t$, and $\mathbf{B}_{t+1} = \mathrm{AdaptOpt}(\tilde{\mathbf{B}}_t, \eta, \nabla\ell(\mathbf{W}_t)^\top \tilde{\mathbf{A}}_t, t)$ with first and
         second moments scaled by $\sqrt{\tilde{s}}$ and $\tilde{s}_t$;
6      **else**
7          Update $\mathbf{A}_{t+1} = \tilde{\mathbf{A}}_t - \eta\nabla\ell(\mathbf{W}_t)\tilde{\mathbf{B}}_t$, $\tilde{\mathbf{B}}_{t+1} = \tilde{\mathbf{B}}_t - \eta\nabla\ell(\mathbf{W}_t)^\top \tilde{\mathbf{A}}_t$;
8      **end**
9 **end**
**Output:** $\mathbf{A}_T$ and $\mathbf{B}_T$.

---

### D.2 Setups for visualization in Figure 1

Figure 1 considers linear regression

$$\min_{\mathbf{W}} \|\mathbf{Y} - \mathbf{W}\mathbf{X}\|_{\mathrm{F}}^2$$

where $\mathbf{X} \in \mathbb{R}^{n \times k}$ and $\mathbf{Y} \in \mathbb{R}^{m \times k}$ are random data matrices with entries generated from standard Gaussian distribution $\mathcal{N}(0, 1)$. The corresponding LoRA objective is

$$\min_{\mathbf{A},\mathbf{B}} \|\mathbf{Y} - (\mathbf{W}^{\mathrm{pt}} + \mathbf{A}\mathbf{B}^\top)\mathbf{X}\|_{\mathrm{F}}^2. \tag{27}$$

For simplicity, we set $m = n = k = 2$, $r = 1$, and $\mathbf{W}^{\mathrm{pt}} = \mathbf{0}$. $\mathbf{A}_0$ and $\mathbf{B}_0$ are randomly initialized from $\mathcal{N}(0, 10)$ and $\mathcal{N}(0, \frac{1}{10})$.

Although it is well-known that linear regression has a closed-form solution through least squares, we consider it here because its Lipschitz smoothness constant can be computed analytically as $L = \|\mathbf{X}\mathbf{X}^\top\|_2$, allowing us to track the upper bound (7).

### D.3 Setups for matrix factorization

Matrix factorization aims to solve

$$\min_{\mathbf{A},\mathbf{B}} \frac{1}{2}\|\mathbf{Y} - \mathbf{A}\mathbf{B}\|_{\mathrm{F}}^2$$

where $\mathbf{Y} \in \mathbb{R}^{m \times n}$ is a given low-rank matrix. It can be viewed as a special case of the linear model (27) with $k = n$, $\mathbf{X} = \mathbf{I}_n$, and $\mathbf{W}^{\mathrm{pt}} = \mathbf{0}$. This test utilizes $m = 128$, $n = 100$, and $r = 8$. The low-rank matrix $\mathbf{Y}$ is generated from standard Gaussian $\mathcal{N}(0,1)$, and then truncated to the largest $r$ singular values. Following the standard LoRA initialization, $\mathbf{A}_0$ is sampled from standard Gaussian $\mathcal{N}(0,1)$, and $\mathbf{B}_0 = \mathbf{0}$. Standard GD is employed in LoRA for $\forall t$, while RefLoRA and ScaledGD are applied when $t > 0$.

### D.4 Details of Datasets

Our evaluations are carried out on commonly-used datasets in the literature.

**GLUE benchmark.** The General Language Understanding Evaluation (GLUE) benchmark is designed to provide a general-purpose evaluation of natural language understanding (NLU) [59]. Those adopted in our work include

- **MNLI** [63] (Multi-Genre Natural Language Inference) tests a model's ability to perform natural language *inference* across different genres of text.
- **SST-2** [53] (Stanford Sentiment Treebank) is a *sentiment analysis* dataset with binary labels.
- **MRPC** [13] (Microsoft Research Paraphrase Corpus) focuses on *paraphrase detection*; i.e. identifying whether two sentences are semantically equivalent.
- **CoLA** [62] (Corpus of Linguistic Acceptability) requires models to judge whether a sentence is *linguistically acceptable*.
- **QNLI** [44] (Question Natural Language Inference) is a question-answering dataset converted to a binary *inference* task.
- **QQP**[2] (Quora Question Pairs) contains pairs of *questions* and the task is to determine if they are semantically equivalent.
- **RTE**[3] (Recognizing Textual Entailment) consists of sentence pairs for textual entailment *inference*.
- **STS-B** [7] (Semantic Textual Similarity Benchmark) evaluates the *textual similarity* of sentence pairs on a continuous scale.

These datasets present a comprehensive benchmark to test general-purpose language models and are distributed under various permissive licenses. A summary of these datasets can be found in Table 5.

Table 5: Summary of GLUE benchmark datasets

| Dataset | Task type | |Train| | |Test| | Metric(s) |
|---|---|---|---|---|
| MNLI | Natural language inference | 393k | 20k | Matched & mismatched accuracies |
| SST-2 | Sentiment analysis | 67k | 1.8k | Accuracy |
| MRPC | Paraphrase detection | 3.7k | 1.7k | Accuracy, F1 |
| CoLA | Acceptability judgment | 8.5k | 1k | Matthews correlation |
| QNLI | QA/NLI | 105k | 5.4k | Accuracy |
| QQP | Paraphrase detection | 364k | 391k | Accuracy, F1 |
| RTE | Textual entailment | 2.5k | 3k | Accuracy |
| STS-B | Semantic similarity | 7k | 1.4k | Pearson & Spearman Correlations |

**Commonsense reasoning.** This category includes tasks that require models to apply everyday knowledge and infer beyond explicit textual information. These datasets are vital for evaluating a model's ability to reason about physical and social contexts. The considered datasets include

- **BoolQ** [10] (Boolean Questions) is a reading comprehension dataset of yes/no questions paired with Wikipedia passages, testing a model's ability to extract and reason over text.
- **WG** [49] (WinoGrande) is a challenging dataset designed to reduce annotation artifacts present in traditional Winograd schemas.

---

[2]`https://quoradata.quora.com/First-Quora-Dataset-Release-Question-Pairs`
[3]`https://paperswithcode.com/dataset/rte`

- **PIQA** [4] (Physical Interaction QA) assesses knowledge of physical commonsense and intuitive physics.
- **SIQA** [50] (SOCIAL-I-QA) focuses on social interaction and social commonsense reasoning.
- **HS** [68] (HellaSwag) aims at evaluating grounded commonsense inference for multiple-choice sentence completion.
- **ARC** [9] (AI2 Reasoning Challenge) contains grade-school level science questions, split into **ARCe** (ARC-easy) and **ARCc** (ARC-challenge), based on difficulty.
- **OpenbookQA** [42] involves multiple-choice science questions that require integrating commonsense and scientific facts.

These datasets are drawn from multiple domains and present diverse reasoning challenges. All datasets used in our work are publicly available under open or research-friendly licenses. Table 6 summarizes these datasets.

Table 6: Summary of commonsense reasoning datasets

| Dataset | Task type | |Train| | |Test| | Metric |
|---|---|---|---|---|
| WinoGrande | Coreference resolution | 40k | 1.3k | Accuracy |
| PIQA | Physical reasoning | 16k | 3k | Accuracy |
| SIQA | Social reasoning | 33k | 2k | Accuracy |
| HellaSwag | Sentence completion | 70k | 10k | Accuracy |
| ARC-easy | Multiple choice QA | 2.3k | 1.2k | Accuracy |
| ARC-challenge | Multiple choice QA | 2.6k | 1.2k | Accuracy |
| OpenbookQA | Open-book QA | 5.0k | 500 | Accuracy |

### D.5 Details on LLMs

We summarize the adopted language models in our evaluation. All model checkpoints are obtained from HuggingFace.

**DeBERTaV3-base** [21] is a transformer-based language model with 184 million parameters. The model checkpoint[4] is released under the MIT license.

**GPT3-turbo** is a proprietary language model accessible via the OpenAI API. While the model weights are not publicly available, its tokenizer[5] is open-sourced under the MIT license.

**LLaMA-7B** [57] is a decoder-only transformer model, which is part of the LLaMA (Large Language Model Meta AI) series. The chekpoint[6] is intended for research use under a non-commercial license.

**LLaMA2-7B** [58] is a refined successor to LLaMA. Its checkpoint[7] is under a permissive license for both research and commercial use.

**LLaMA3-8B** [17] is part of the third generation of LLaMA series. The checkpoint[8] is released under a permissive Meta license for both research and commercial applications.

Stable Diffusion V1.4 [45] is a latent text-to-image diffusion model released by CompVis, Stability AI, and Runway. The checkpoint[9] is made available under the CreativeML-OpenRAIL-M license.

### D.6 Hyperparameters for fine-tuning LLMs

**GLUE** setup follows from [23, 70], where LoRA is applied to all linear modules. The results for Full FT, BitFit, Adapters, LoRA, DoRA and AdaLoRA in Table 2 are taken from [70], while the remaining results are obtained from our experiments. We fix the LoRA rank as $r = 8$ and scaling

---

[4] https://huggingface.co/microsoft/deberta-v3-base

[5] https://huggingface.co/Xenova/gpt-3.5-turbo

[6] https://huggingface.co/huggyllama/llama-7b

[7] https://huggingface.co/meta-llama/Llama-2-7b

[8] https://huggingface.co/meta-llama/Meta-Llama-3-8B

[9] https://huggingface.co/CompVis/stable-diffusion-v1-4

factor as $\alpha = 8$, and search the optimal learning rate from $\eta \in \{1 \times 10^{-3}, 8 \times 10^{-4}, 4 \times 10^{-4}\}$. Batch size is fixed as 32 for all datasets. The default AdamW [40] optimizer and linear learning rate scheduler are utilized for all tests. Other hyperparameters are gathered in Table 7. Note that RefLoRA requires less fine-tuning epochs compared other LoRA variants due to its fast convergence.

Table 7: Hyperparameters for GLUE benchmark

| Dataset | $\eta$ | Epochs | Warmup steps | Max seq. len. | Cls. dropout | Weight decay |
|---------|--------|--------|--------------|---------------|--------------|--------------|
| MNLI | $4 \times 10^{-4}$ | 5 | 1000 | 256 | 0.15 | 0 |
| SST-2 | $1 \times 10^{-3}$ | 2 | 500 | 128 | 0 | 0.01 |
| MRPC | $4 \times 10^{-4}$ | 10 | 50 | 128 | 0 | 0.01 |
| CoLA | $1 \times 10^{-3}$ | 5 | 100 | 64 | 0.15 | 0 |
| QNLI | $4 \times 10^{-4}$ | 5 | 500 | 512 | 0.1 | 0.01 |
| QQP | $1 \times 10^{-3}$ | 5 | 1000 | 320 | 0.2 | 0.01 |
| RTE | $8 \times 10^{-4}$ | 10 | 50 | 320 | 0.2 | 0.01 |
| STS-B | $1 \times 10^{-3}$ | 10 | 100 | 128 | 0.1 | 0.1 |

**Commonsense reasoning** setup is from [24, 65], where LoRA is attached to linear projections in transformers' self-attention and feedforward modules. The results for ChatGPT, LoRA, and DoRA in Table 3 are taken from [65], while the remaining are acquired through our tests. We test with LoRA ranks $r \in \{16, 32\}$ and scaling factor fixed as $\alpha = 2r$. The learning rate is tuned from $\eta \in \{8 \times 10^{-5}, 1 \times 10^{-4}, 2 \times 10^{-4}, 3 \times 10^{-4}\}$. The tuned learning rate per model can be found in Table 8. The number of fine-tuning epochs is set to 2, and batch size is 16 for all tests. The default AdamW [40] optimizer and linear learning rate scheduler are utilized with 100 warmup steps. Dropout rate is 0.05. The remaining hyperparameters are set to the default values used in [24].

Table 8: Learning rates for LLaMA models

| | LLaMA-7B | | LLaMA2-7B | | LLaMA3-8B | |
|---|---|---|---|---|---|---|
| Rank $r$ | 16 | 32 | 16 | 32 | 16 | 32 |
| Learning rate $\eta$ | $2 \times 10^{-4}$ | $2 \times 10^{-4}$ | $3 \times 10^{-4}$ | $2 \times 10^{-4}$ | $8 \times 10^{-5}$ | $1 \times 10^{-4}$ |

We also attempted to include LoRA-Pro [61]; however, it incurred runtime costs that exceeded the limits of our resources. Similar scalability concerns have been reported by other users in the community; see e.g., issue #6 on the LoRA-Pro GitHub repository. For this reasons, we omitted it from our final results.

**Subject-driven image generation** utilizes the default setups in [48]. Specifically, LoRA is applied to the `to_k`, `to_q`, `to_v`, `to_out`, `add_k_proj`, and `add_v_proj` modules of U-net with rank $r = 4$, and scaling factor $\alpha = 4$. The diffusion model is fine-tuned with batch size 1 and learning rate $\eta = 10^{-4}$ for 500 iterations. AdamW with 0.01 weight decay is adopted as the optimizer.

# E    Scaling to larger models

Scalability does not confine the applicability of RefLoRA. In larger models, the major runtime bottleneck is the forward pass and gradient computation. In comparison, the additional overhead of RefLoRA becomes negligible. Table 9 compares the computational overheads of LoRA variants under various model sizes, where gradient checkpointing is turned on for Gemma3-27B-pt so that the model can fit within a single NVIDIA H100 96GB GPU. Notably, the runtime and memory gap between LoRA and RefLoRA narrows as model size increases. These findings indicate that RefLoRA remains as practical and efficient as LoRA, especially for larger models.

Table 9: Throughput (it/s↑) and GPU memory consumption (GB↓) under various models sizes.

| Method | DeBERTaV3-base | | LLaMA3-8B | | Gemma3-27B-pt | |
|---|---|---|---|---|---|---|
| | Tp. | Mem. | Tp. | Mem. | Tp. | Mem. |
| LoRA | $1\times$ (2.26) | 8.73 | $1\times$ (1.53) | 36.21 | $1\times$ (0.58) | 64.28 |
| LoRA-RITE | $0.73\times$ | 8.87 | $0.63\times$ | 37.37 | $0.95\times$ | 64.28 |
| RefLoRA | $0.88\times$ | 8.86 | $0.87\times$ | 36.35 | $0.98\times$ | 64.28 |
| RefLoRA-S | $0.99\times$ | 8.73 | $0.99\times$ | 36.21 | $0.99\times$ | 64.28 |

