# OpenReview forum: "RefLoRA: Refactored Low-Rank Adaptation for Efficient Fine-Tuning of Large Models"
_NeurIPS.cc/2025/Conference — NeurIPS 2025 poster_

### Official Review · Reviewer_USmE · 2025-06-27

**Clarity:** 3
**Significance:** 3
**Originality:** 3
**Rating:** 4
**Confidence:** 4

**Summary:**

The paper “RefLoRA: Refactored Low-Rank Adaptation for Efficient Fine-Tuning of Large Models” improves upon standard LoRA by dynamically re‐factorizing its low‐rank updates to minimize a provable upper bound on the loss, yielding more balanced and consistent weight adjustments. Empirical results show that RefLoRA and its lightweight variant converge faster and more stably on low‐rank matrix problems and standard PEFT benchmarks.

**Questions:**

Questions:
1. Can the authors provide a convergence theorem, similar to [1] (Theorem 2)? I think the preconditioning of the gradients are somewhat related to the manifold optimization problem introduced in [1]. I.e. they combat the problem of high curvature of the manifold consisting of rank r matrices. Can the authors comment on this?
2. What happens if the Gram matrices A^TA and B^TB are near singular? Computation of $\tilde{S}$ can become hard?
3. Have you evaluated performing the refactor step every k>1 iterations (instead of every step)? How does this affect convergence speed and runtime?

Comments:
1. Assumption 1 can also be enforced by using the methods of [1,2].
2. In [1], the authors also discuss the shortcomings of the LoRA learning dynamics and how to circumvent them. It is relevant work that should be added to the manuscript.




[1] https://arxiv.org/abs/2410.18720
[2] https://arxiv.org/abs/2205.13571

**Ethical Concerns:**

["NO or VERY MINOR ethics concerns only"]

**Final Justification:**

The authors have addressed my concerns and thus I have raised the quality score of my review.
I think my current score (4) reflects the work well, since there are some theoretical shortcomings about convergence that other methods have figured out.
The empirical results of the paper and the presented analysis however, warrant acceptance.

**Limitations:**

- Ill conditioning of the Gram matrices needs to be addressed.

**Quality:**

4

**Strengths And Weaknesses:**

Strength:

-  The method derivation is sold - I did not spot any logic flaws.
- Pseudocode is well readable.
- I appreciate the matrix completion benchmark - as it gives good insight in the method without the bloat of a full LLM architecture

Weakness:

- It is a bit unclear how the simplified method (Algorithm 2) is derived.
- See questions below

---

> ### Author Rebuttal · Authors · 2025-07-28
>
> The detailed comments and suggestions are greatly appreciated. Next, we deal with the concerns one-by-one below.
>
> 1. **Derivation of the lightweight RefLoRA-S (Algorithm 2)**
> Thanks for pointing this out. We will elaborate further in the main context to improve clarity.
> The principle for deriving RefLoRA-S is the same as RefLoRA. But instead of finding an $r \times r$ matrix $\mathbf{S}_ t$, RefLoRA-S aims at a scalar $s_ t \in \mathbb{R}_ {++}$ for computational efficiency. In particular, we replace $\mathbf{S}_ t = s_ t \mathbf{I}_ r$ in Eq. (8), which results in $
> \min_{s_t > 0} ( || \mathbf{A}_ t ||_ \mathrm{F}^2 s_ t + || \mathbf{B}_ t ||_\mathrm{F}^2 s_t^{-1} - \frac{1}{L \eta} )^2$; cf. Eq. (21) in Appendix B.4. More details for solving this quartic minimization problem can be also found in the Appendix.
>
> 2. **Convergence theorem, and gradient preconditioning in relation to manifold optimization problem**
> Thanks for providing the nice intuition and relevant work [1]. We will make sure to touch upon it in the revised draft.
> Agreed, RefLoRA is related to manifold optimization at high level. Yet, a similar derivation of [1, Theorem 2] is non-straightforward, as the manifold in RefLoRA is less structural.
> In particular, GeoLoRA mainly works with two (generalized) Stiefel/Grassmanian manifolds, whereas the manifold of RefLoRA can be interpreted as $\{(\tilde{\mathbf{A}}, \tilde{\mathbf{B}}) | {\cal L}(\mathbf{A}, \mathbf{B})  = {\cal L}(\tilde{\mathbf{A}}, \tilde{\mathbf{B}})\}$ for given $\mathbf{A}, \mathbf{B}$.
> Because $\mathbf{A}$ and $\mathbf{B}$ are updated over iterations, this "manifold" is time-varying and data-dependent. As a tradeoff, working with such an irregular manifold reduces the number of retractions compare to GeoLoRA.
> While we are still working toward convergence analysis for the general case, the convergence of RefLoRA in the simpler matrix factorization case can be indeed established by adapting the proof of [3, Theorem 1.1]. Using the notation in [3], we redefine $A := (\tilde{U} + \tilde{V}) / 2$, and $B := (\tilde{U} - \tilde{V}) / 2$, where $\tilde{U}$ and $\tilde{V}$ are the correspondents of $U$ and $V$ after refactoring. A key step in the original proof involves bounding the asymmetry $|| B ||_F$, ensuring it remains sufficiently small after stage one and throughout stage two. This condition is inherently satisfied under our refactoring. As a result, it can be verified that the induction hypothesis in [3] remains valid under RefLoRA.
>
> 3. **Convergence and runtime of refactoring per $k$ steps**
> We conducted additional ablation tests under the same configuration described in Section 4.4; and term the suggested variant as RefLoRA-I, for which the refactoring interval is set to $k = 10$ so that its throughput matches RefLoRA-S.
> The following two tables respectively outline the finetuning loss over epochs, and the computational overheads comparison. The first table shows that the convergence of RefLoRA-I is slightly slower than RefLoRA, but comparable to RefLoRA-S. This is because the refactoring in RefLoRA not only balances the norms of $\mathbf{A}_t$ and $\mathbf{B}_t$, but also rotates and skews them to align their Gram matrices. While the norms of $\mathbf{A}_t$ and $\mathbf{B}_t$ evolve gradually due to the small learning rate, their Gram matrices can diverge significantly after gradient updates, necessitating more frequent adjustment to maintain stability. Furthermore, despite RefLoRA-I with $k=10$ matches RefLoRA-S in throughput, it consumes as much memory as RefLoRA. Thus, RefLoRA-S remains preferable in terms of both time and space efficiency.
>
>  Epoch | LoRA | LoRA-Pro | LoRA-RITE | RefLoRA | RefLoRA-S | RefLoRA-I
> :---:|:---:|:---:|:---:|:---:|:---:|:---:
>  2 | 0.444 | 0.410 | 0.442 | 0.316 | 0.340 | 0.335
>  4 | 0.248 | 0.238 | 0.160 | 0.082 | 0.091 | 0.124
>  6 | 0.174 | 0.164 | 0.147 | 0.041 | 0.056 | 0.062
>  8 | 0.151 | 0.115 | 0.065 | 0.006 | 0.040 | 0.040
>  10 | 0.084 | 0.065 | 0.042 | 0.007 | 0.025 | 0.031
>
>  Method | Throughput (it/s) | Memory (GB)
> :---:|:---:|:---:
>  LoRA | 1$\times$ (2.26) | 8.73
>  LoRA-RITE | 0.73$\times$ | 8.87
>  RefLoRA | 0.88$\times$ | 8.86
>  RefLoRA-S | 0.99$\times$ | 8.73
>  RefLoRA-I | 0.99$\times$ | 8.86
>
> 4. **Remedy for nearly singular Gram matrices**
> To ensure numerical stability, a small $\epsilon$ has been added to the eigenvalues of $\mathbf{A}^\top \mathbf{A}$ when calculating its inverse; see lines 66, 71 and 107 of codes `reflora.py`. This is a standard practice in optimization to prevent division by zero. The same technique is employed in optimizers including AdamW, LoRA-RITE, and LoRA-Pro.
>
>
> 5. **Related works [1,2]**
> Thank you for bringing these two papers to our attention. They are closely related to our work, and the manifold perspective offers a complementary lens to interpret our results. We will incorporate a discussion of both studies in the revised submission.
>
> [1] S. Schotthöfer, E. Zangrando, G. Ceruti, F. Tudisco, and J. Kusch, "GeoLoRA: Geometric integration for parameter efficient fine-tuning," *arXiv preprint*, 2024.
> [2] S. Schotthöfer, E. Zangrando, J. Kusch, G. Ceruti, and F. Tudisco, "Low-rank lottery tickets: finding efficient low-rank neural networks via matrix differential equations," *arXiv preprint*, 2022.
> [3] T. Ye, and S. Du, "Global convergence of gradient descent for asymmetric low-rank matrix
> factorization," in *NeurIPS*, 2021.

---

> ### Comment · Reviewer_USmE · 2025-08-03
>
> Thank you for the answers.
>
> > While we are still working toward convergence analysis for the general case, the convergence of RefLoRA in the simpler matrix factorization case can be indeed established by adapting the proof of [3, Theorem 1.1]
>
> If I understand [3] correctly, they consider a **matrix approximation problem**, i.e.,
> $$
> \min_{U, V} \|\Sigma - UV\|_2,
> $$
> where $\Sigma$ is a low-rank matrix.
>
> How would the result transfer to the **neural network training problem** of
> $$
> \min_{U, V} \mathcal{L},
> $$
> where $U, V$ are parametrizations of a non-linear neural network, and $\mathcal{L}$ is a more general (and possibly non-convex) loss function?

---

> ### Author Response · Authors · 2025-08-03
>
> Thank you for the prompt reply, and we apologize for not clearly motivating the matrix approximation setting.
>
> We mentioned the matrix approximation problem in our previous reply because it is among the simplest LoRA scenarios, i.e., applying LoRA to a single-layer neural network with whitened data [4]. This setting and its variants have recently been adopted as a prototypical case for analyzing LoRA [5,6,7]: although nonconvex (in U and V), it admits global convergence guarantees in polynomial time. We take this case as a preparation toward broader settings, including those highlighted by the reviewer.
>
> Regarding extensions to general loss functions, additional work is needed because analysis tools in [1] do not directly apply under our manifold structure. Our current approach connects the analysis to continuous time: with careful initialization and a small modification, RefLoRA closely relates to a gradient flow. The continuous-time limit reduces to establishing convergence of this flow, which can be handled with standard techniques. We are now analyzing its discretization; once the discretization error is controlled, the corresponding convergence guarantees for RefLoRA follow directly.
>
> Should any aspects remain unclear, we would be glad to discuss more. Thank you!
>
> [4] S. Arora, N. Cohen, N. Golowich, W. Hu, "A convergence analysis of gradient descent for deep linear neural networks," ICLR, 2019.
>
> [5] F. Zhang, M. Pilanci, "Riemannian preconditioned lora for fine-tuning foundation models," ICML, 2024.
>
> [6] B. Li, L. Zhang, A. Mokhtari, N. He, "On the crucial role of initialization for matrix factorization," ICLR, 2025.
>
> [7] Y. Zhang, F. Liu, Y. Chen, "LoRA-One: one-step full gradient could suffice for fine-tuning large language models, provably and efficiently," ICML 2025.

---

> > ### Comment · Reviewer_USmE · 2025-08-05
> >
> > I thank the authors for their answers. I will keep my score.

---

> > > ### Author Response · Authors · 2025-08-05
> > >
> > > We sincerely appreciate the time and effort you have devoted to helping us improve our manuscript. If any questions or concerns still remain, we would be glad to provide further clarification.

---

### Official Review · Reviewer_r7FQ · 2025-07-01

**Clarity:** 4
**Significance:** 3
**Originality:** 4
**Rating:** 5
**Confidence:** 3

**Summary:**

The paper introduces RefLoRA, a novel method for efficient fine-tuning of large language models (LLMs) by addressing the limitations of Low-Rank Adaptation (LoRA). The authors propose dynamically selecting the optimal low-rank factorization per step to minimize an upper bound on the loss, leading to balanced updates and faster convergence. The method is evaluated on matrix factorization, natural language understanding, and commonsense reasoning tasks, demonstrating improved performance over existing LoRA variants. The theoretical analysis is rigorous, and the experimental results are comprehensive, covering multiple LLMs.

**Questions:**

1. The largest model tested is LLaMA3-8B. The authors mention scaling to 30B models as future work, but can RefLoRA’s overhead become prohibitive for even larger models (e.g., 100B+ parameters)?
2. RefLoRA closes the gap to full fine-tuning, but the gap persists in some tasks. Is this due to inherent limitations of low-rank updates, or could further refinements help?

**Ethical Concerns:**

["NO or VERY MINOR ethics concerns only"]

**Limitations:**

Yes.

**Quality:**

3

**Strengths And Weaknesses:**

Strengths:
1. The paper provides a solid theoretical foundation for RefLoRA, including proofs for optimal factorization and balanced updates (Section 3). The derivation of the closed-form solution for the optimal S_t (Theorem 3) is particularly compelling.
2. The experiments are extensive, covering diverse tasks and models. The results consistently show RefLoRA’s superiority over LoRA and other variants.
3. The paper is well-written, with clear motivations, methods, and results. The visualization of loss landscapes (Figure 1) effectively supports the theoretical claims.
Weaknesses:
1. While RefLoRA-S is introduced as a simplified variant, its trade-offs (e.g., performance vs. complexity) are not thoroughly explored. For instance, how much performance is sacrificed compared to full RefLoRA, especially in low-rank settings (r=16)?
2. The GLUE results for LoRA-Pro and LoRA-RITE are omitted due to runtime or convergence issues. This raises questions about the fairness of comparisons.

---

> ### Author Rebuttal · Authors · 2025-07-28
>
> We appreciate the reviewer for the valuable feedback provided. The questions are addressed one-by-one below.
>
> 1. **Performance-complexity trade-off of RefLoRA-S**
> There is indeed a tradeoff. The performance comparison of RefLoRA and RefLoRA-S were listed in Table 3 of the original submission. Specifically, RefLoRA achieves 0.09% higher average accuracy than RefLoRA-S across three models in the high-rank $r=32$ setting, while the gap increases to 0.23% in the low-rank $r=16$ setup. This trend is because a higher rank provides sufficient model expressiveness, allowing the loss to be minimized effectively even with scalar refactoring. In contrast, at lower rank $r=16$, RefLoRA-S exhibits slower convergence compared to RefLoRA. These observations suggest that full refactoring becomes more critical when the number of finetuning parameters is limited.
> In terms of runtime and memory, RefLoRA introduces additional computational overhead of $\mathcal{O}((m+n+r)r^2)$ time and $\mathcal{O}(r^2)$ space, which grows rapidly with $r$. For instance, when finetuning LLaMA3-8B with $r=32$, RefLoRA's throughput is 12.1% lower than RefLoRA-S, and consumes additional 0.14 GB GPU memory; see the table below for details. These considerations point to a practical trade-off: RefLoRA-S is particularly preferable for larger $r$, offering improved throughput and memory usage, while solely incurring a marginal drop in performance.
>
> 2. **Comparison with LoRA-Pro and LoRA-RITE**
> Please refer to our "Response 2" to Reviewer VpnN, where additional numerical tests are conducted for the requested comparison.
>
> 3. **Scaling to 100B+ models**
> Scalability does not confine the applicability of RefLoRA. In larger models, the major runtime bottleneck is the forward pass and gradient computation. In comparison, the additional overhead of RefLoRA becomes negligible.
> Limited by computational resources, we are unable to perform experiments on models with 100B+ parameters. Instead, it is possible to scale up to Gemma3-27B-pt, where gradient checkpointing is turned on so that the model can fit within a single NVIDIA H100 96GB GPU. The computational comparison is summarized in the table below. Notably, the runtime and memory gap between LoRA and RefLoRA narrows as model size increases. These findings indicate that RefLoRA remains as practical and efficient as LoRA, especially for larger models.
>  Method | DeBERTaV3 |-base | LLaMA3 |-8B | Gemma3 |-27B |
> :---:|:---:|:---:|:---:|:---:|:---:|:---:
>  | | Tp. (it/s) | Mem. (GB) | Tp. (it/s) | Mem. (GB) | Tp. (it/s) | Mem. (GB)
>  LoRA | 1$\times$ (2.26) | 8.73 | 1$\times$ (1.53) | 36.21 | 1$\times$ (0.58) | 64.28
>  LoRA-RITE | 0.73$\times$ | 8.87 | 0.63$\times$ | 37.37 | 0.95$\times$ | 64.28
>  RefLoRA | 0.88$\times$ | 8.86 | 0.87$\times$ | 36.35 | 0.98$\times$ | 64.28
>  RefLoRA-S | 0.99$\times$ | 8.73 | 0.99$\times$ | 36.21 | 0.99$\times$ | 64.28
>
> 4. **Gap relative to full fine-tuning**
> The performance of RefLoRA is still limited by the low-rank premise of LoRA. However, the small gap observed is a reasonable tradeoff, given that the primary motivation behind LoRA is to fuel scenarios where full fine-tuning is infeasible.
> To further fill this gap, a promising direction is to integrate RefLoRA with recent advancements that support high-rank update, such as ReLoRA [1] and HiRA [2].
> [1] V. Lialin, S. Muckatira, N. Shivagunde, and A. Rumshisky, "ReLoRA: High-Rank Training Through Low-Rank Updates," in *ICLR*, 2024.
> [2] Q. Huang, T. Ko, Z. Zhuang, L. Tang, and T. Zhang, "HiRA: Parameter-Efficient Hadamard High-Rank Adaptation for Large Language Models," in *ICLR*, 2025.

---

> > ### Author Response · Authors · 2025-08-05
> >
> > Thank you once again for your insightful review and interest in our work! If you have any remaining questions or concerns, we would be happy to provide further clarification. We sincerely appreciate your time and consideration!

---

### Official Review · Reviewer_VpnN · 2025-07-04

**Clarity:** 2
**Significance:** 3
**Originality:** 3
**Rating:** 4
**Confidence:** 3

**Summary:**

The paper introduces RefLoRA, a novel approach to improve Low-Rank Adaptation (LoRA) for parameter-efficient fine-tuning (PEFT) of large language models (LLMs). It addresses LoRA’s limitations, such as slow convergence and unbalanced weight updates due to nonunique low-rank factorizations, by optimizing the factorization process to minimize a loss upper bound, resulting in faster and more stable convergence.

**Questions:**

RefLoRA is tested only on transformer-based LLMs (DeBERTaV3, LLaMA series) for NLU and reasoning tasks. Can the authors provide evidence or analysis of RefLoRA’s applicability to other architectures (e.g., vision transformers, diffusion models) or tasks (e.g., text generation, RLHF)?
The paper omits LoRA-Pro and LoRA-RITE due to runtime and convergence issues (Section D.6). Can the authors provide a partial comparison, alternative evaluation, or explanation of why these methods failed, and how RefLoRA compares to other recent LoRA variants?

**Ethical Concerns:**

["NO or VERY MINOR ethics concerns only"]

**Limitations:**

Yes.

**Quality:**

2

**Strengths And Weaknesses:**

The strengths of this paper are as follows.
Rigorous Theory: RefLoRA’s optimization of the SPD matrix $ \mathbf{S}_t $ is supported by detailed proofs (Appendix B, Theorem 3), ensuring mathematical rigor.
Strong Experiments: Outperforms LoRA, DoRA, and AdaLoRA on GLUE (88.50 vs. 88.15) and reasoning tasks (78.85 vs. 76.19) using DeBERTaV3 and LLaMA models (Tables 2–3).

There are some weaknesses in this paper:
Limited Model Scope: Tested only on transformer-based LLMs, limiting generalizability (not addressed in limitations).
Incomplete Baselines: Omits LoRA-Pro and LoRA-RITE due to runtime/convergence issues (Section D.6), weakening claims.

---

> ### Author Rebuttal · Authors · 2025-07-28
>
> We thank the reviewer for the thorough comments and insightful suggestions. Our responses to the concerns raised are given next.
>
> 1. **Applicability to other model architectures and tasks**
> Following the reviewer's suggestion, further numerical tests are conducted on a subject-driven image generation task [1] with Stable Diffusion v1.4. The goal is to finetune a diffusion model using a few user-provided images so that it can generate the same object in various contexts. Specifically, the model is finetuned on a set of images labeled "a photo of sks dog," and subsequently evaluated by generating images under the prompt "a sks dog eating nachos." LoRA adapters with rank $r=4$ are attached to the U-Net component of Stable Diffusion, and experimental setups including hyperparameters are default to those in [1].
> The finetuning losses for LoRA, LoRA-Pro, LoRA-RITE, and RefLoRA are summarized in the table below, where average and standard deviation are calculated over 3 random runs. It is seen that RefLoRA achieves 14.0% , 13.1%, and 9.5% improvements over LoRA and LoRA-Pro, and  LoRA-RITE, respectively. In addition to quantitative gains in loss, we also observe noticeable improvements in image quality. The generations from RefLoRA-tuned models show markedly clearer details and better object fidelity, particularly in the mouth and tongue regions, where features are often distorted in outputs from other three baselines.
> As external links are not permitted, the corresponding visual results will be included in the revised manuscript. We are also pleased to provide them during rebuttal if rule-compliant methods become available.
>  Loss ($\downarrow$) | LoRA | LoRA-Pro | LoRA-RITE | RefLoRA
> :---:|:---:|:---:|:---:|:---:
>  Avg$\pm$std | $0.100 \pm 0.015$ | $0.099 \pm 0.015$ | $0.095 \pm 0.016$ | $0.086 \pm 0.017$
>
> [1] N. Ruiz, Y. Li, V. Jampani, Y. Pritch, M. Rubinstein, K. Aberman, "DreamBooth: Fine Tuning Text-to-Image Diffusion Models for Subject-Driven Generation," in *CVPR*, 2023.
>
> 2. **Comparison with LoRA-Pro and LoRA-RITE**
> We encountered practical challenges when scaling LoRA-Pro to larger LLaMA models with 7B and 8B parameters. For instance, the estimated time for finetuning a LLaMA3-8B model with LoRA-Pro on two NVIDIA A100 GPUs required more than one day, which exceeded our available computational budget. Similar scalability concerns have been reported by other users in the community; see e.g., issue #6 on the LoRA-Pro GitHub repository. We hope to revisit these experiments once performance optimizations are available.
> As for LoRA-RITE, our initial experiments encountered some instability and sensitivity due to random seeds. Fortunately, the codebase has since been updated, and a stable version was released recently. Using the latest implementation, we obtained stable and reliable results. As shown in the table below, RefLoRA(-S) consistently outperforms LoRA-RITE across all three backbone models, showcasing its robustness and effectiveness. Additionally, RefLoRA(-S) also demonstrates better computational efficiency. For example, when finetuning LLaMA3-8B, the throughput of LoRA-RITE reaches merely 72.4% (63.6%) of RefLoRA(-S), yet consuming an additional 1.02 GB (1.16 GB) GPU memory. Please refer to the table in our response to Reviewer r7FQ for further details.
>  Model | Method | BoolQ | PIQA | SIQA | HS | WG | ARCe | ARCc | OBQA | Avg
> :---:|:---:|:---:|:---:|:---:|:---:|:---:|:---:|:---:|:---:|:---:
>  LLaMA-7B | LoRA | 66.42 | 80.03 | 77.84 | 82.88 | 81.85 | 79.92 | 63.40 | 77.20 | 76.19
>  LLaMA-7B | LoRA-RITE | 69.82 | 82.75 | 78.55 | 84.72 | 81.69 | 82.15 | 66.23 | 81.40 | 78.54
>  LLaMA-7B | RefLoRA | 69.60 | 82.48 | 79.53 | 88.25 | 82.56 | 81.57 | 66.64 | 80.20 | **78.85**
>  LLaMA-7B | RefLoRA-S | 70.18 | 82.48 | 78.15 | 87.41 | 82.08 | 81.52 | 65.36 | 81.60 | 78.60
>  LLaMA2-7B | LoRA | 69.8 | 79.9 | 79.5 | 83.6 | 82.6 | 79.8 | 64.7 | 81.0 | 77.6
>  LLaMA2-7B | LoRA-RITE | 71.04 | 82.43 | 79.79 | 89.12 | 84.53 | 83.88 | 68.77 | 81.20 | 80.10
>  LLaMA2-7B | RefLoRA | 72.54 | 83.79 | 80.04 | 86.94 | 84.85 | 86.36 | 71.50 | 80.20 | **80.78**
>  LLaMA2-7B | RefLoRA-S | 73.36 | 83.84 | 80.76 | 90.02 | 82.48 | 84.55 | 67.92 | 82.60 | 80.69
>  LLaMA3-8B | LoRA | 70.8 | 85.2 | 79.9 | 91.7 | 84.3 | 84.2 | 71.2 | 79.0 | 80.8
>  LLaMA3-8B | LoRA-RITE | 74.19 | 89.44 | 81.52 | 95.44 | 86.74 | 90.45 | 80.12 | 86.60 | 85.56
>  LLaMA3-8B | RefLoRA | 75.35 | 88.74 | 80.91 | 95.71 | 86.66 | 90.49 | 80.20 | 87.40 | 85.68
>  LLaMA3-8B | RefLoRA-S | 75.50 | 89.72 | 81.11 | 95.59 | 87.29 | 90.99 | 79.78 | 86.00 | **85.75**

---

> > ### Author Response · Authors · 2025-08-05
> >
> > Thank you again for your thoughtful review and engagement with our work! If you have any remaining questions or concerns, we would be happy to clarify further. If all your concerns have been adequately addressed, please kindly consider updating the score to reflect your current assessment. We appreciate your time and consideration!

---

### Decision · Program_Chairs · 2025-09-17

**Decision:**

Accept (poster)

**Comment:**

The paper received mixed ratings initially. There are a few minor concerns from the reviewers. The rebuttal has addressed most of them, and the reviewers raised their scores. The final decision is acceptance.